# An iPSC-derived astrocyte model of fragile X syndrome exhibits dysregulated cholesterol homeostasis

Karo Talvio[1], Victoria A. Wagner [2], Rimante Minkeviciene[1], Jay S. Kirkwood[3], Anna O. Kulinich[2], Juzoh Umemori[4], Anil Bhatia[3], Manhoi Hur [3], Reijo Käkelä [5], Iryna M. Ethell [2] & Maija L. Castrén [1✉]

Cholesterol is an essential membrane structural component and steroid hormone precursor, and is involved in numerous signaling processes. Astrocytes regulate brain cholesterol homeostasis and they supply cholesterol to the needs of neurons. ATP-binding cassette transporter A1 (ABCA1) is the main cholesterol efflux transporter in astrocytes. Here we show dysregulated cholesterol homeostasis in astrocytes generated from human induced pluripotent stem cells (iPSCs) derived from males with fragile X syndrome (FXS), which is the most common cause of inherited intellectual disability. ABCA1 levels are reduced in FXS human and mouse astrocytes when compared with controls. Accumulation of cholesterol associates with increased desmosterol and polyunsaturated phospholipids in the lipidome of FXS mouse astrocytes. Abnormal astrocytic responses to cytokine exposure together with altered anti-inflammatory and cytokine profiles of human FXS astrocyte secretome suggest contribution of inflammatory factors to altered cholesterol homeostasis. Our results demonstrate changes of astrocytic lipid metabolism, which can critically regulate membrane properties and affect cholesterol transport in FXS astrocytes, providing target for therapy in FXS.

[1] Department of Physiology, Faculty of Medicine, University of Helsinki, Helsinki, Finland. [2] Division of Biomedical Sciences, and Neuroscience Graduate Program, School of Medicine, University of California Riverside, Riverside, CA, USA. [3] Metabolomics Core Facility, Institute for Integrative Genome Biology, University of California Riverside, Riverside, CA, USA. [4] Gene and Cell Technology, A.I.Virtanen Institute, University of Eastern Finland, Kuopio, Finland. [5] Helsinki University Lipidomics Unit, HiLIPID, Helsinki Institute of Life Science, HiLIFE, Biocenter Finland (Metabolomics), and Molecular and Integrative Biosciences Research Programme, Faculty of Biological and Environmental Sciences, University of Helsinki, Helsinki, Finland. ✉email: maija.castren@helsinki.fi

Cholesterol is an essential structural component of biological membranes and a precursor of steroid hormones[1]. It is crucial for synapse formation and function as high cholesterol content is required in lipid rafts[2]. Interaction of cholesterol with a number of ion channels can facilitate or inhibit channel function and contribute to the regulation of neuronal activity[3,4]. The cellular distribution of cholesterol is determined by its affinity for sphingolipids and aversion to highly unsaturated phospholipids (PLs)[5]. These molecular interactions promote the segregation of membrane microdomains regarded crucial for the functions of many integral proteins and their interactions[1]. The brain cholesterol pool is separated from other cholesterol pools of the body and both neurons and astrocytes synthesize cholesterol in the brain[6]. Astrocytes produce excess cholesterol and secrete most of the produced cholesterol for use by neurons[7]. The main cholesterol efflux transporter in astrocytes is ATP-binding cassette transporter A1 (ABCA1), which transfers cellular cholesterol and PLs onto lipid-poor apolipoproteins that shuttle cholesterol[8–10].

Cholesterol homeostasis is dysregulated in many syndromes associated with autism spectrum disorder (ASD), including fragile X syndrome (FXS)[11]. FXS is the most common inherited intellectual disability syndrome and monogenic cause of ASD[12]. FXS is caused by the absence of Fragile X messenger ribonucleoprotein (FMRP) which is essential for normal synapse formation and plasticity[13]. Treatment with lovastatin, an inhibitor of 3-hydroxy-3-methylglutarylcoenzyme A (HMG-CoA) reductase which regulates the early irreversible and rate-limiting step in the biosynthesis of cholesterol, dampens neuronal hyperexcitability in the brain of FXS mouse model, Fmr1 KO mice, and rescues part of the mouse FXS phenotype[14]. Early brief treatment with lovastatin also prevented associative learning deficits in Fmr1 KO rats[15], but cholesterol was not measured in relation to lovastatin treatment responses in the rodent FXS models.

Lovastatin treatment was found to improve the daily living skills and communication subscales in individuals with FXS in the first open-label study[16]. The treatment effects were less, but nonetheless visible particularly when lovastatin was combined with minocycline also in another pilot study of 22 participants[17]. However, lovastatin did not enhance the treatment response of parent-implemented language intervention in a randomized, double-blind trial[18], and it remained unclear whether lovastatin has less effect on speech-related mechanisms than mechanisms underlying other symptoms such as hyperactivity and stereotypy in FXS. In addition, individual differences in FXS and the combined use of other drugs may influence the outcome of clinical trials with lovastatin[16]. Individuals with FXS have reduced peripheral cholesterol and triacylglycerol (TAG), and an abnormal profile of fatty acids (FA)[19–21], which questions the appropriateness of the lovastatin treatment. Brain region-specific alterations in cholesterol metabolism in a rat model of FXS have been shown by using enzymatic colorimetric cholesterol assay[22]. Furthermore, FMRP deficiency has an impact on lipid homeostasis[23] and n-3 polyunsaturated fatty acid (n-3 PUFA) supplementation has beneficial effects on the behavioral phenotype of the Fmr1 KO mice[24,25].

Given the crucial role of astrocytes in the maintenance of cholesterol homeostasis in brain, we investigated cholesterol balance in FXS astrocytes. We found that the expression of ABCA1 was reduced in FMRP-deficient astrocytes derived from human induced pluripotent stem cells (iPSCs) generated from FXS males and from the cortices of Fmr1 KO mice. The concomitantly increased astrocytic cholesterol and desmosterol levels were associated with membrane restructuring that favored polyunsaturated PL species at the expense of sphingomyelin (SM) and phosphatidylcholine (PC) species with a low degree of unsaturation. We also explored the role of the pro-inflammatory status of the FXS astrocytes in the reduced ABCA1 expression and observed dysregulated cytokine/chemokine secretome profile and FXS astrocyte reactivity in response to cytokines. We suggest that pre-inflammatory conditions and altered cholesterol homeostasis may impede FXS astrocytes affecting glia-neuronal interactions in FXS.

## Results

**Reduced ABCA1 expression in human FXS astrocytes**. ABCA1 transfers cellular cholesterol and PLs onto lipid-poor apolipoproteins that shuttle esterified cholesterol[9,10]. We found that ABCA1 mRNA expression was reduced ($P = 0.011$) in human forebrain astrocytes generated from three FXS patient-specific male iPSC lines compared with controls (Fig. 1a). Similarly, human embryonic stem cell (hESC)-derived FMR1 KO astrocytes expressed less ABCA1 than their isogenic controls (log2 −6.41-fold change, $P = 0.0004$, P adjust = 0.125) in RNA-Seq analysis[26]. ABCA1 protein expression was also reduced in FXS iPSC-derived astrocyte cultures ($P = 0.013$; Fig. 1b, c).

**Maintenance of normal extracellular cholesterol levels in human FXS astrocyte cultures**. Astrocytes produce cholesterol on demand of neurons and secrete most of their cholesterol[7]. To assess the impact of reduced astrocytic cholesterol transporter expression on cholesterol secretion from FXS astrocytes, we compared total cholesterol content in astrocyte conditioned medium (ACM) collected from serum-free human astrocyte cultures derived from four FXS and three control iPSC lines. We observed that cholesterol content did not differ between FXS and control ACM (Fig. 1d). Exposure to retinoic acid (RA), which can increase cholesterol secretion through activation of liver X receptor (LXR)-ABCA1 signaling, did not measurably affect extracellular cholesterol levels in FXS or control astrocyte cultures (Fig. 1d).

Utilizing mass spectrometry (MS), we found that cholesterol levels were not significantly different between FXS and control ACM ($P = 0.093$), although there was a trend suggesting an increase (Fig. 1e). There is evidence that human astrocytes secrete large cholesterol-rich lipoproteins and most secreted cholesterol is esterified (cholesteryl ester, CE), unlike mainly non-esterified cholesterol containing lipoproteins secreted from mouse astrocytes[27]. The CE amounts did not differ between human FXS and control samples under basal conditions or after treatment with RA ($P = 0.409$ and $0.945$, respectively; Fig. 1f). However, the ratio of cholesterol to CE was higher in FXS secretomes than in controls ($P = 0.029$; Fig. 1g).

The stored and secreted CE are formed from cholesterol and long-chain fatty acyl-CoA[28]. FA transport protein FAT1P encoded by the SLC27A1 gene has acyl-CoA synthetase activity and plays a crucial role in the uptake of long-chain FAs across the plasma membrane[29]. Similar to ABCA1 expression, SLC27A1 was reduced in FXS iPSC-derived astrocytes (Fig. 1h) when compared with controls. Downregulated FATP1 could affect lipid balance by reducing uptake of long-chain FAs in FXS astrocytes.

**ABCA1-related changes in conditioned medium of Fmr1 KO astrocytes**. The FXS-specificity of the reduced ABCA1 expression in human forebrain FXS astrocyte model was confirmed by the observation of reduced ABCA1 immunoreactivity in cortical astrocytes of Fmr1 KO mice compared with wild type (WT) controls ($P = 0.034$; Fig. 2a). As in human ACM, the levels of cholesterol ($P = 0.634$; Fig. 2b) and CE ($P = 0.202$; Fig. 2c) did not differ in ACM from Fmr1 KO astrocytes compared to WT, using MS. Unexpectedly, a substantial decrease in desmosterol (367.3357 m/z; $P = 0.030$) (Fig. 2d), the last precursor of cholesterol in the Bloch unsaturated sterol side chain pathway[30], was seen in the positive ion mode in Fmr1 KO ACM when compared to that in WT ACM.

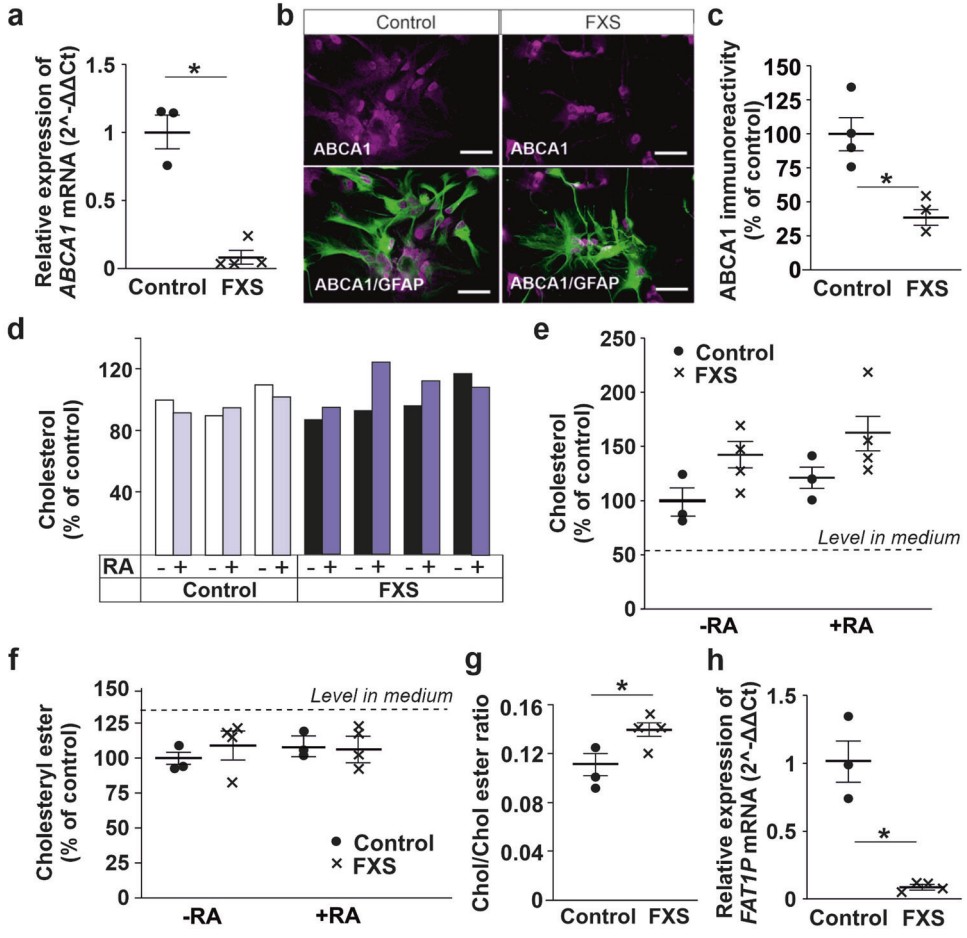

**Fig. 1 Expression of cholesterol efflux transporter ABCA1 and extracellular cholesterol in human iPSC-derived FXS and control astrocyte cultures.**
**a** Expression of *ABCA1* mRNA in human control and FXS astrocytes; $n$(Control) = 3 cell lines and $n$(FXS) = 4 cell lines (3 biologically independent cell lines). **b** Immunofluorescence images showing ABCA1 (red) and GFAP (green) in control and FXS astrocytes, scale bar 50 μm. **c** Relative ABCA1 protein expression in control and FXS astrocytes; $n$(Control) = 241 cells and $n$(FXS) = 285 cells on 3–4 cultures/each cell line (4 control and 3 FXS cell lines). **d** Relative abundance of total cholesterol in astrocyte-conditioned medium (ACM) of control (HEL23.3, HEL24.3, and HEL46.11 cell lines from left to right, respectively) and FXS (HEL69.6, HEL70.3, HEL70.6, and HEL100.2 cell lines from left to right, respectively) astrocytes under basal conditions and after retinoic acid (RA) treatment. **e** Relative abundance of cholesterol and **f** cholesteryl esters in control and FXS ACM in LS-MS-based lipidomics analysis; $n$(Control) = 3 cell lines and $n$(FXS) = 4 (3 biologically independent cell lines). Dashed line shows cholesterol (**e**) and cholesteryl ester (**f**) levels in cell-free medium. **g** Ratio of cholesterol/cholesteryl ester relative abundance in control and FXS ACM under basal condition. **h** Expression of *FAT1P* in human control and FXS astrocytes; $n$(Control) = 3 cell lines and $n$(FXS) = 4 (3 biologically independent cell lines). Data are shown as mean ± SEM. The two-sided *t*-test was applied to evaluate the statistical differences; *$P < 0.05$.

In addition to cholesterol, ABCA1 exports PC, phosphatidylserine (PS), and SM from the cytoplasmic to the exocytoplasmic leaflet of membranes, and its ATPase activity is particularly stimulated by PC, PS, and SM[31]. We found that lipid profiles were different between ACM from *Fmr1* KO and WT mouse astrocytes and that the differences appeared to be the most pronounced in the profiles of PC species (Fig. 2e). Levels of monounsaturated PC species (e.g., 32:1 and 34:1) were lower and there were slight elevations of polyunsaturated PC species in *Fmr1* KO ACM compared with controls (Fig. 2f, Supplementary Figure 1). These PC differences were also apparent in the PCA analysis (Fig. 2g). The altered PC profile of *Fmr1* KO ACM suggested that the absence of FMRP led to dysregulation of ABCA1-mediated efflux of PC along with cholesterol from *Fmr1* KO astrocytes.

**Cholesterol accumulation and altered lipidome of *Fmr1* KO mouse astrocytes.** We found that reduced ABCA1 is associated with several changes in the lipidome of *Fmr1* KO astrocytes in

mass spectrometry analysis. Cholesterol abundance was increased in *Fmr1* KO astrocytes when compared with WT controls ($P = 0.038$; Fig. 3a). Since total cholesterol in human and mouse astrocyte medium was not affected by FMRP deficiency, the data suggested accumulation of cholesterol in astrocytes. However, increased desmosterol in KO astrocytes ($P = 0.036$; Fig. 3b) concomitant with reduced desmosterol abundance in ACM suggested higher uptake of desmosterol by *Fmr1* KO mouse astrocytes in the serum-containing medium. As desmosterol is a precursor that is needed for cholesterol synthesis, the data likely reflected higher cholesterol synthesis by *Fmr1* KO mouse astrocytes. Desmosterol accumulation could also be regarded as an attempt to activate LXR signaling and thereby increase sterol secretion from astrocytes[32,33].

In addition, we found that the profiles of membrane PL species in *Fmr1* KO and WT astrocytes were different (Fig. 3c). Comparison of the PL species composition of *Fmr1* KO and WT astrocytes using mass spectrometry revealed that KO astrocytes had increased levels of polyunsaturated PC,

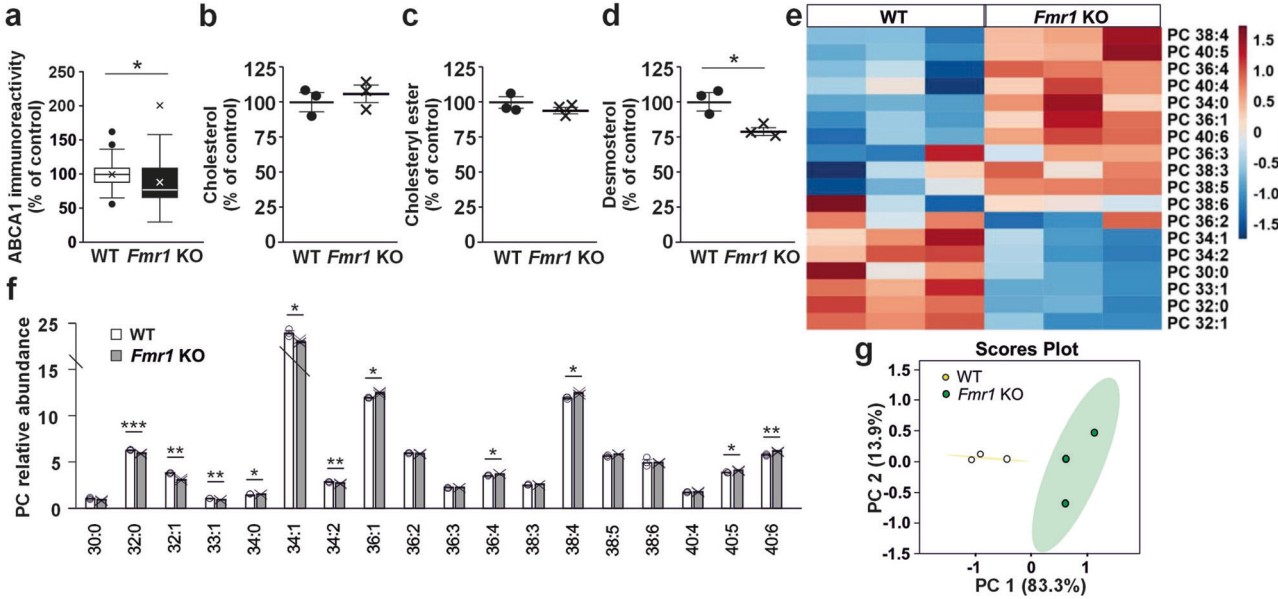

**Fig. 2 Lipid changes in *Fmr1* KO astrocyte conditioned medium. a** Relative ABCA1 immunoreactivity in cultured mouse WT and *Fmr1* KO astrocytes. Data are mean ± SEM; $n = 48$–57 images, 4–5 astrocyte cultures per group isolated from P1 WT and KO littermates. **b** Cholesterol relative abundance, **c** cholesteryl ester relative abundance, and **d** desmosterol relative abundance in WT and *Fmr1* KO mouse astrocyte conditioned medium (ACM) in LS-MS-based lipidomics analysis. **e** Heat map (data shown as z scores) showing profile of phosphatidylcholine (PC) species in WT and *Fmr1* KO ACM in LC-MS analysis. **f** Bar graph of PC species profiles also indicating the values for individual samples and **g** principal component analysis (PCA) scores plot of the PC species in WT and *Fmr1* KO ACM. **b–g** Data of mouse ACM; mean ± SEM, $n$(WT) = 3 and $n$(*Fmr1* KO) = 3 biologically independent astrocyte cultures. The $t$ test was applied to evaluate the statistical differences; *$P < 0.05$, **$P < 0.01$, ***$P < 0.001$. Soft Independent Modeling of Class Analogies (SIMCA) confirmed statistically significant differences in PC species composition in the WT and *Fmr1* KO medium samples ($P < 0.05$).

phosphatidylethanolamine (PE), and phosphatidylserine (PS) species and some species with the shortest chains, and contained less PC and SM harboring one or two monounsaturated acyl chains (Fig. 3d–f). The PC species profile was also significantly different between the WT and KO astrocytes (Fig. 3e). Consistent with lipidomic findings in ACM, the polyunsaturated species showed higher levels in KO astrocytes, likely affecting vesiculation, membrane fluidity, and lipid-derived mediator signaling that is involved in a variety of biological processes in concert with cytokines. While the total amount of PS species was increased in KO cells, the SM levels were decreased (Fig. 3f). The phosphatidylinositol (PI) levels remained constant and PS/PI ratio was higher in KO than in WT astrocytes (Fig. 3g, h). Our findings suggested accumulation of cholesterol and increased polyunsaturated membrane PLs in *Fmr1* KO astrocytes.

**Altered cytokine responses in FXS astrocytes.** Since ABCA1 activity is implicated in neuroinflammation, and inflammatory factors are involved in the regulation of its expression[34], we explored pro-inflammatory mechanisms, which could contribute to reduced *ABCA1* expression and altered cholesterol homeostasis in FXS astrocytes. Using a commercial cytokine antibody array, we found that several anti-inflammatory cytokines were reduced in ACM of human FXS astrocytes. Interleukin-13 (IL-13) and IL-10 were the most reduced cytokines in FXS astrocytes compared with controls (Fig. 4a). Since there is evidence that IL-13 stimulates ABCA1 expression[35], its low level (54.3% of control) could contribute to the reduced ABCA1 expression in FXS astrocytes. On the other hand, reduced IL-10 (54.3% of control) may be the consequence of the low ABCA1 expression[36]. In contrast, we did not find similar clear differences in pro-inflammatory cytokines IL-1α, IL-1β, IL-6, IL-8, and TNF-α between the secretome from control and FXS astrocytes (Fig. 4a). Of the CC chemokine ligands in the array, CCL1 expression was abnormally low (36.2% of

control) in FXS ACM compared to controls in repeated experiments (Fig. 4a). CCL1 triggers chemotaxis of Th2 and a subset of T regulatory cells[37], and its immunomodulatory function has been shown to be regulated by statins[38]. RANTES (CCL5) was previously identified among reduced chemokines in plasma of FXS individuals[39], but its extracellular level (95.4% of control) (Fig. 4a) or cellular mRNA expression ($P = 0.688$) was not different between FXS and control astrocytes under basal conditions (Fig. 4b). However, the expression of *CCL5* was sixfold higher ($P < 0.0001$) in FXS astrocytes activated with IL-1β (10 ng/ml) and 2-fold lower ($P < 0.0001$) after the treatment with TNF-α (50 ng/ml) compared to controls (Fig. 4c), indicating abnormalities in responses to pro-inflammatory cytokines[40] in FXS astrocytes. Analogously, treatment with IL-1β resulted in sixfold increase ($P < 0.0001$) in *GFAP* expression, whereas TNF-α treatment decreased (twofold) *GFAP* expression in FXS astrocytes compared with controls (Fig. 4d).

## Discussion

The brain contains ~25% of total cholesterol in the body[7]. Astrocytes comprise the largest cell population involved in the regulation of cholesterol homeostasis which is critical for the developing brain. Both astrocytes and neurons synthesize cholesterol in the developing brain, but unlike neurons, astrocytes do not normally accumulate cholesterol and they secrete most of their newly synthesized cholesterol[7]. Balance between cholesterol secretion and cellular retention is tightly regulated to avoid undesirable toxic effects of excess cholesterol in the extracellular space[41]. Cholesterol-induced toxicity has been identified as a critical factor in the pathogenesis of several diseases leading to interest in the therapeutic use of cholesterol-lowering medications[14–18,41]. In the present study, effects of the absence of FMRP on astrocytic cholesterol balance were studied in a cell type-dependent manner in human and mouse FXS astrocytes. We

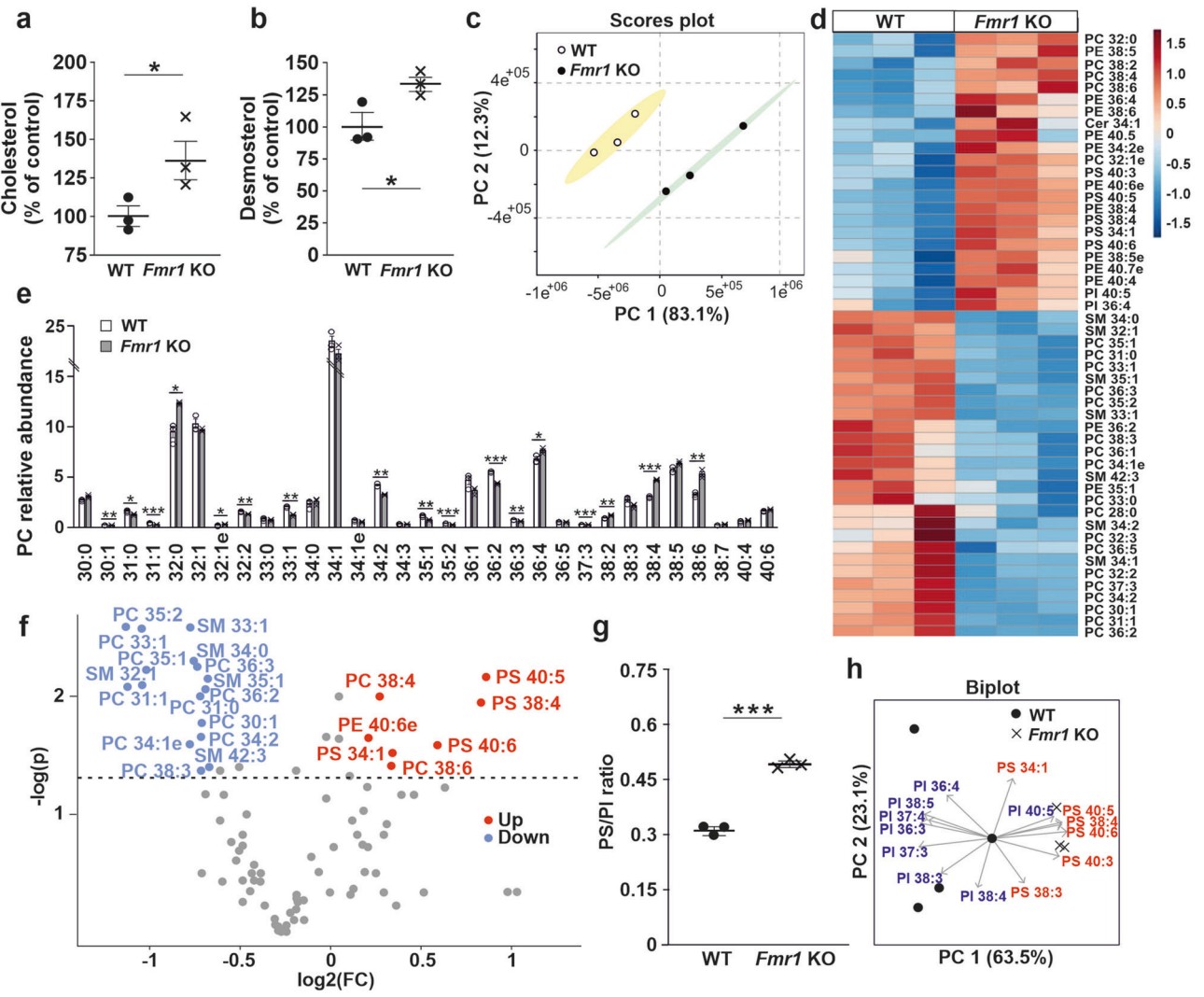

**Fig. 3 Lipidome of *Fmr1* KO mouse astrocytes. a** Cholesterol and **b** desmosterol relative abundance in WT and *Fmr1* KO mouse astrocytes. **c** Principal component analysis (PCA) scores plot of membrane phospholipid species in WT and *Fmr1* KO mouse astrocytes. **d** Heat map of membrane lipids (data shown as z scores) showing increased levels of polyunsaturated phospholipid species and also some species with the shortest chains, but poor in phospholipids harboring one or two monounsaturated acyl chains in WT controls and *Fmr1* KO astrocytes. **e** Phosphatidylcholine (PC) species profiles in WT and *Fmr1* KO astrocytes, also indicating the values for individual samples. **f** Volcano plot of the membrane phospholipid species that were down (blue) or up (red) in *Fmr1* KO compared to WT astrocytes. **g** Ratio of membrane "acidic" lipids phosphatidylserine (PS) and phosphatidylinositol (PI) in WT and *Fmr1* KO astrocytes. **h** PCA biplot of phosphatidylserine (PS) and phosphatidylinositol (PI) species in WT and *Fmr1* KO astrocytes. All data were generated in LC-MS lipidomics analysis; $n$(WT) = 3 and $n$(*Fmr1* KO) = 3 biologically independent astrocyte cultures. Data are shown as mean ± SEM. *$P < 0.05$, **$P < 0.01$, ***$P < 0.001$. Soft Independent Modeling of Class Analogies (SIMCA) confirmed that the WT and *Fmr1* KO astrocyte samples had statistically significantly different phospholipid species, PC species, and acidic phospholipid (PS and PI) species compositions ($P < 0.05$).

showed that dysregulated cholesterol homeostasis in FXS astrocytes manifested as decreased cholesterol transporter ABCA1 expression both in human and mouse astrocytes. In human FXS astrocytes, the ABCA1 defect was associated with changes in cytokine secretion, while in mouse *Fmr1* KO astrocytes cultured with serum, the ABCA1 defect caused accumulation of cholesterol and desmosterol connected with changes in inflammatory properties and lipidome (Fig. 5).

ABCA1 is a transmembrane protein, which displays a crucial function in the regulation of cholesterol balance by mediating the efflux of intracellular free cholesterol and PLs across the plasma membrane[10]. Astrocytes produce cholesterol on demand of neurons and ABCA1 is the main cholesterol efflux transporter in astrocytes[10,31]. To avoid the toxicity of cholesterol, its content has to be accurately controlled. We found that extracellular cholesterol content was maintained normally in cultures of human and

mouse FXS astrocytes expressing reduced levels of ABCA1. Decrease in ABCA1 led to the accumulation of astrocytic cholesterol, suggesting activation of a non-ABCA1-mediated efflux or increase in diffusion, perhaps due to a larger cholesterol gradient between medium lipoproteins and cellular plasma membrane[42].

Cellular PC level and species composition affect intracellular distribution, transport, and ABCA1-mediated secretion of cholesterol[28]. The PC species profile was different between the control and *Fmr1* KO astrocytes, and polyunsaturated PC species were elevated both in *Fmr1* KO ACM and in lysates from KO astrocytes compared to the controls. PC is the most abundant PL class in eukaryotic cells and thus serves as a cholesterol sink that buffers cells against cholesterol-induced ER stress[43]. PC also serves as a precursor in SM synthesis required for membrane nanodomains enriched with SM and cholesterol. In KO astrocytes, the membrane PL profile favored the highly unsaturated PC

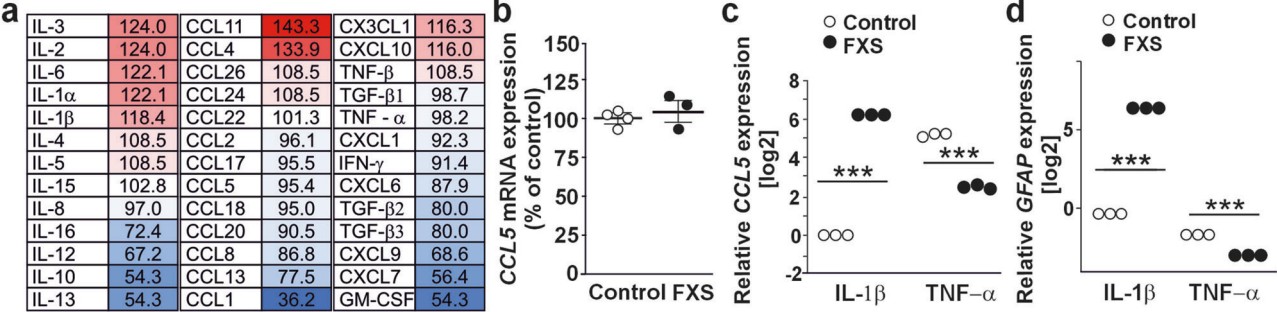

**Fig. 4 Pre-inflammatory changes of human iPSC-derived FXS astrocytes. a** Comparison of cytokine and chemokine profile in screening of human forebrain astrocytes derived from FXS and control human iPSCs. Values are percentages for FXS, calculated against the control in pooled four control and four FXS iPSC-derived ACM. **b** Expression of *CCL5* mRNA in control and FXS astrocytes using *GAPDH* expression for normalization. $n$(Control) = 3 and $n$(FXS) = 3 cell lines representing biological variants. **c** *CCL5* and **d** *GFAP* mRNA expression in control and FXS astrocytes relative to untreated controls after treatment with IL-1β and TNF-α. Astrocytes were generated from control (HEL11.4) and FXS (HEL69.5) human iPSC lines and treated with IL-1β (10 ng/ml) and TNF-α (50 ng/ml) for 7 days. mRNA levels were analyzed by RT-qPCR with three technical replicates using the ΔΔCt method. Data are mean ± SEM. ***$P < 0.001$ by two-way ANOVA.

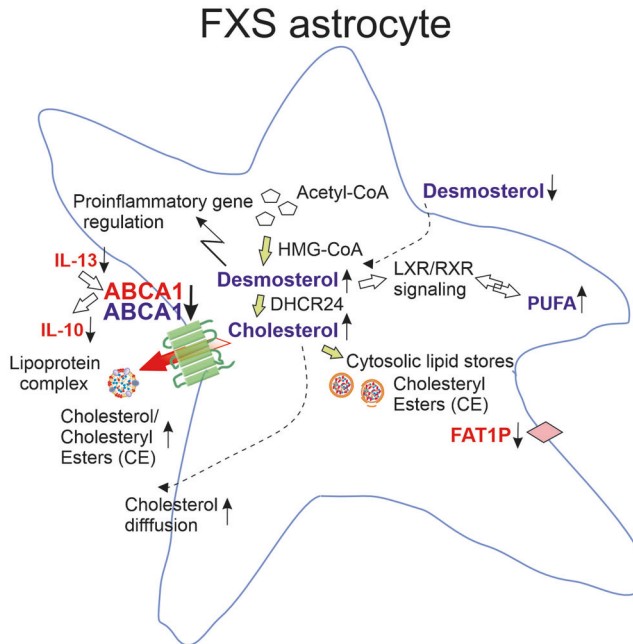

**Fig. 5 Graphical presentation summarizing mechanisms involved in dysregulated cholesterol homeostasis in human and mouse FXS astrocytes.** Studies of human forebrain FXS astrocytes revealed reduced ABCA1 and fatty acid (FA) transporter *FAT1P* expression, which is associated with an increase in cholesterol/cholesteryl ester ratio in astrocyte-conditioned medium (ACM). The levels of IL-13 and IL-10 involved in the regulation of ABCA1 expression were reduced in ACM of human FXS astrocytes. Reduced ABCA1 protein in *Fmr1* KO astrocytes was combined with increased cholesterol and desmosterol, while desmosterol was decreased in *Fmr1* KO ACM, suggesting altered LXR/RXR signaling that contributed to the increased polyunsaturated fatty acid (PUFA) content in FXS astrocytes. Changes observed in human FXS and mouse *Fmr1* KO astrocytes are shown in red and blue, respectively.

species and contained less SM, rendering the membrane structurally incompatible with cholesterol[44]. Thus, the capacity of the KO astrocyte membranes to buffer cholesterol overload and prevent cholesterol toxicity was compromised.

The high levels of polyunsaturated PL species in the KO astrocytes likely had several consequences. High PL unsaturation

may modify membrane physical properties and augment release of CE and TAG-containing vesicles from ER[45,46]. In addition, increased availability of PUFA ligand for nuclear receptor LXR alters LXR-mediated gene expression[47]. LXR/retinoic X receptor (RXR) ligands are known to stimulate ABCA1 expression and reduced ABCA1 in FXS astrocytes may reflect defective LXR/RXR signaling potentially caused by dysregulated RXRA and RXRB by the absence of FMRP[48]. *Fmr1* KO astrocytes also displayed abnormally high level of desmosterol, which is the second most abundant sterol in the brain and implicated in the regulation of synaptic plasticity[30]. It has important signaling roles and as an LXR agonist it inhibits SREBP target genes and selectively reprograms fatty acid metabolism in a cell type-specific manner[49,50]. Since desmosterol impairs membrane lipid packing compared to cholesterol, it could contribute to membrane fluidity control in cells with excess cholesterol[51].

In addition to LXR-mediated effects, desmosterol may have LXR-independent oxidative and inflammatory effects[49], whereas PUFAs function as precursors of different lipid mediators required to control inflammation[52]. We observed that secreted anti-inflammatory cytokines were low in human FXS ACM. Particularly the levels of interleukins connected to the regulation of ABCA1 expression and cholesterol homeostasis were low. It is possible that low levels of IL-13 contributed to the reduced ABCA1 expression in FXS astrocytes[35]. Respectively, low ABCA1 expression might have been involved in reduced IL-10 in FXS ACM[36]. A deficit of IL-10 has also been reported in the *Fmr1* KO hippocampus[24]. Impaired immune responses in FXS astrocytes are consistent with previously found dysregulation of several pro-inflammatory cytokines in *Fmr1* KO mice[24] and in clinical studies of FXS[39,53]. Abnormal astrocytic inflammatory IL-6 response has been associated with synaptic dysfunction in *Fmr1* KO mice[54].

Our finding that *Fmr1* KO astrocytes had high PS content and low relative amounts of saturated and monounsaturated SM species is especially interesting. Serine is the common precursor for PS and SM synthesis, and thus, in the KO astrocytes lipid synthesis likely favored pathways producing PS instead of SM. High membrane PS activates many membrane-bound kinases, which have been studied as targets of therapy in FXS[55]. In addition, elevated PS to SM ratio, as in the KO astrocytes, activates protein kinase C by positive co-operativity[56]. Protein kinase C activation promotes apoA-I-induced stabilization of ABCA1 protein[57], downregulated in the KO astrocytes. Despite increased PS in KO astrocytes, the PI levels remained constant. Numerous signaling functions utilize

phosphorylated PIs, which are also involved in the regulation of intracellular transport of cholesterol at membrane contact sites, e.g., between the ER and Golgi complex[58]. Apparently, the abundant PI precursor supported such transport in FXS astrocytes.

Statin treatment might improve behavioral outcomes in FXS children[16,17] and lovastatin has beneficial effects on the phenotype of *Fmr1* KO mice[14]. Lovastatin treatment prevented susceptibility to audiogenic seizures and excessive hippocampal protein synthesis in these mice. The treatment dampened neuronal hyperexcitability caused by hypersensitivity to mGluR5-ERK1/2 activation[59]. However, since protein synthesis inhibitors were not able to produce these effects and the response was not replicated by treatment with simvastatin[60], mechanisms conferring lovastatin-mediated beneficial effects on the FXS phenotype remained unclear. Our study demonstrated that altered cholesterol homeostasis in FXS astrocytes was associated with broader changes in lipidome and inflammatory factors, which are potentially directly modulated by statins[61]. The study also suggested that the increase in polyunsaturated membrane PLs is a coplayer in impairing the FXS astrocyte cholesterol metabolism. Clinical studies have shown that peripheral cholesterol levels are reduced in FXS males and the recent report of Abolghasemi et al. showed an abnormal plasma lipid profile including decreased n-6 PUFAs and n-3 PUFAs in FXS[21]. Among the n-3 PUFAs, eicosapentaenoic acid (EPA, 20:5n-3) and docosahexaenoic acid (DHA, 22:6n-3) were the most reduced. Furthermore, n-3 PUFA supplementation of *Fmr1* KO mice from weaning to adulthood rescues many behavioral symptoms and neuroinflammatory parameters[24]. Several clinical trials have shown beneficial effects of n-3 PUFA supplementation in ASD and ADHD, but not all results are consistent and the ratio of EPA and DHA in n-3 PUFA supplements and the overall dietary intake of FAs may affect the clinical outcome[62–65]. For example, excess EPA may functionally inhibit ABCA1-mediated cholesterol efflux[66]. There is also evidence that EPA and DHA modulate differentially cytokine expression[67] The monocyte IL-10 expression, found to be reduced in FXS astrocytes, was higher when the cells were supplemented with EPA than with DHA. Potential detrimental effects of altering the optimal ratio of EPA to DHA on cognitive functions have been reported[68].

Lipids have fundamental roles in astrocyte function, including energy metabolism, maintaining membrane lipid-protein interactions, and cell-to-cell signaling. Cholesterol is a major lipid component of the plasma membrane of cells and it is essential in the maintenance of membrane fluidity, thickness, and segregation of the lipid domains necessary for multiple signaling platforms. Cholesterol is co-localized with SM and PC in the plasma membranes and at the surface of lipoprotein particles. The present study demonstrated that FMRP-deficiency in astrocytes affects the balance of the membrane lipids, decreasing the proportion of PC and SM species with a low degree of unsaturation, which is structurally compatible with cholesterol, and increasing the proportion of polyunsaturated PL species, incompatible with cholesterol but having crucial signaling roles. In addition to inhibiting ABCA1 function in cholesterol efflux, the biased membrane lipid composition potentially interferes with functions of several channels on the membrane directly or by altering bilayer stiffness and hydrophobic mismatch between transmembrane domains and the lipid bilayer[69]. Our results, first, highlight the role of ABCA1 in the regulation of cholesterol in FXS astrocytes, and encourage further studies to explore ABCA1, which targeting for treatment may provide beneficial effects in FXS. Second, the observed increases in polyunsaturated PLs in the *Fmr1* KO astrocytes indicate a need for further studies on the regulation of membrane PL profile to develop novel treatments using the human iPSC-derived astrocytes model.

## Methods

**Human astrocytes**. Human forebrain astrocytes were generated as described previously[70] from male FXS (HEL69.6, HEL70.3, HEL70.6, and HEL100.2) and control (HEL23.3, HEL24.3, HEL46.11, PO2/UEF-3A, and PO4/UEF-3B) iPSC lines[71–75]. In addition, astrocytes were differentiated from the male *FMR1* KO hESC line carrying an FXS-causing mutation and from its isogenic control (H1) hESCs[76]. Informed consent was obtained from all donors of cells used for reprogramming. The research using human cells was approved by the Ethics Committee of the Hospital District of Helsinki and Uusimaa, Finland.

Briefly, human pluripotent stem cells (hPSCs) were maintained on Matrigel-coated plates in Essential 8 medium (E8; Thermo Fisher Scientific Ltd., Vantaa, Finland). For differentiation, 80% confluent hPSCs were dissociated with 0.5 mM ethylenediaminetetraacetic acid (EDTA; Invitrogen, Carlsbad, CA, USA) and plated on a low-attachment six-well plate in E8 with 20 ng/ml human basic fibroblast growth factor (bFGF2; Peprotech, Somerset County, NJ, USA) and 20 μM Rho kinase inhibitor (Sigma-Aldrich, St. Louis, MO, USA). On the following day, the medium was changed to neuronal induction medium [Advanced Dulbecco´s Modified Eagle Medium (DMEM)/F12, 1× N2, 2 mM L-glutamine, 1× non-essential amino acids (NEAA), and 1× penicillin-streptomycin (P-S) (all from Thermo Fisher Scientific)] supplemented with 0.1 μM LDN-193189 (Stemgent), 1 μM cyclopamine (Sigma-Aldrich), 10 μM SB-431542 (Sigma-Aldrich), and 0.5 μg/ml DKK1 (Dickkopf-related protein 1; Peprotech). Advanced DMEM/F12 allows serum reduction due to the added ethanolamine, glutathione, ascorbic acid, insulin, transferrin, AlbuMAX™ II lipid-rich bovine serum albumin (BSA) for cell culture, and the trace elements sodium selenite, ammonium metavanadate, cupric sulfate, and manganese chloride. At 12 days in vitro (DIV), the medium was replaced by neuronal induction medium supplemented with 20 ng/ml brain-derived neurotrophic factor (BDNF; Peprotech), and at 30 DIV by neurosphere medium (Advanced DMEM/F12, 1× B27 -RA, 1× L-glutamate, 1× NEAA, 1× penicillin–streptomycin (P-S) [all from Thermo Fisher Scientific]) supplemented with 20 ng/ml bFGF2 and 20 ng/ml epidermal growth factor (Peprotech). Growth factors were added three times a week and spheres were dissociated approximately once per week manually. At 60 DIV, spheres were dissociated and progenitors plated on poly-ornithine/laminin-coated (Sigma-Aldrich) culture plates in neurosphere medium supplemented with 20 ng/ml ciliary neurotrophic factor (CNTF; Peprotech). Cells were passaged with Trypsin-EDTA (0,05%) (Thermo Fisher Scientific) approximately once per week and seeded at 20,000 cells/cm². After 75 DIV, cells had acquired astrocyte morphology and they were maintained on Matrigel-coated culture plates. The absence of FMRP in FXS iPSC-derived astrocytes was confirmed in our previous studies. All cell cultures were tested regularly to be free from mycoplasma contamination using a Mycoplasma detection kit (MycoAlert, Lonza Group Ltd.)

**Immunocytochemistry of human astrocytes**. For immunostaining of human astrocytes, cells plated 30,000 cells/well on laminin/poly-L-ornithine (Sigma-Aldrich)-coated coverslips in 24-well plates were fixed with 4% paraformaldehyde for 15 min at room temperature (RT) and washed four times with phosphate-buffered saline (PBS). Permeabilization of cells with 2% BSA, 5% sucrose, 5% skimmed milk, and 0.1% Triton X-100 in PBS for 30 min was followed by incubation with primary antibody overnight at 4°C and with secondary antibody for 3 h at RT in PBS supplemented with 2% BSA, 5% sucrose, and 0.05% Triton X-100. Primary antibodies were chicken anti-GFAP (Abcam, ab#4674, 1:500) and mouse anti-ABCA1 (Abcam, ab#18180, 1:500). Secondary antibodies were anti-chicken Alexa Fluor 488 and anti-mouse Alexa Fluor 647 (both from Invitrogen, 1:1000). Nuclei were stained with 4´6-diamino-2-phenylindole (DAPI; Sigma, D9542, 1:10000) and mounted with Shandon Immuno-Mount (Thermo Scientific).

The stained cells were scanned on the Pannoramic 250 (Flash III) scanner (3D Histech, Budapest, Hungary) with 20X objective (NA 0.8) and sMOS PCO.edge 5.5 2MP camera. The images were analyzed using ImageJ. Intensity threshold was set at 10/255 to differentiate cells from background. Automated particle analysis tool was used to calculate the area and mean intensity in each imaged cell. Area threshold was set at greater than 500 pixels. Mean intracellular intensity was determined and corrected by subtracting the background intensity.

**Mice**. Male *Fmr1* KO and wild-type mice on a congenic C57BL/6 background (The Jackson Laboratory) were used at postnatal day 1 (P1). All studies were performed according to the National Institutes of Health and Institutional Animal Care and Use Committee at the University of California Riverside guidelines. Mice were maintained in an American Association for Accreditation of Laboratory Animal Care (AAALAC) accredited facility under a standard 12 h light/dark cycle and given ad libitum access to food and water.

**Primary mouse astrocytes**. Primary cultures of cortical astrocytes were prepared from the neocortices of P1 *Fmr1* KO and wild-type mice as previously described[77]. Mice were briefly anesthetized by cooling on ice and cleansed with ethanol. Brains were dissected and placed in ice-cold PBS supplemented with 25 mM glucose and 0.1% bovine serum albumin (PGB). Cortices were dissected and meninges removed. Brain tissue was incubated in Papain (0.5 mg/ml) -DNAse (10 μl/ml)

**Table 1 Primers used for qPCR analysis.**

| Gene | Forward primer (5′–3′) | Reverse primer (5′–3′) |
|---|---|---|
| Human *ABCA1* | GCACTGAGGAAGATGCTGAAA | AGTTCCTGGAAGGTCTTGTTCAC |
| Human *CCL5* | CTGCTTTGCCTACATTGCCC | TCGGGTGACAAAGACGACTG |
| Human *GAPDH* | TGTTCCAATATGATTCCACCC | CTTCTCCATGGTGCGTGAAGA |
| Human *GFAP* | ACCTGCAGATTCGAGAAACCAG | GGTCCTGCCTCACATCACATC |
| Human *SLC27A1* | TGACAGTCGTCCTCCGCAAGAA | CTTCAGCAGGTAGCGGCAGATC |

Accession numbers of sequences used to design the primers: AF275948 (human *ABCA1* mRNA); NM_002985 (human *CCL5* mRNA); JO2642 (human *GAPDH* mRNA); NM_002055 (human *GFAP* mRNA); NM_198580 XM_32251 (human *SLC27A1* mRNA).

PGB at 37 °C for 20 min. Tissue pieces were collected by centrifugation for 5 min at $300 \times g$ and gently triturated. Cells were plated in T25 tissue culture flasks prepared by coating for 1–2 h at 37 °C with 50 µg/mL poly-D-lysine at a concentration ~$0.5 \times 10^6$ cells/ml in DMEM supplemented with 10% fetal bovine serum (FBS) and 1× P-S (all from Thermo Fisher Scientific). Astrocytes were grown in a humidified 10% $CO_2$ incubator for 1 week to confluence and adherent microglia was removed by shaking at 180 rpm for 2 h and 240 rpm for 30 min followed by media change. Cells were detached using Trypsin-EDTA (0,05%) (Thermo Fisher Scientific) and plated in the density of $1 \times 10^6$ cells on coated T25 flasks. Astrocyte culture medium was DMEM supplemented with 10% FBS and 1× P-S (all from Thermo Fisher Scientific) and the medium was changed every 2–3 days. Mouse astrocytes were used for liquid chromatography-mass spectrometry (LC-MC) lipidomics experiments at 24 DIV.

**Immunocytochemistry of mouse astrocytes.** For immunostaining, mouse astrocytes were plated on poly-D-lysine coated glass coverslips at the density of $5 \times 10^4$ cells in 24-well plates. After two days at 25 days in vitro (DIV), astrocytes were fixed with 2% paraformaldehyde for 30 min at RT and washed three times with PBS. Cells were permeabilized with 0.1% Triton X-100 in PBS for 30 min and blocked with 1% normal donkey serum for 1 h at RT. Cells were incubated with primary antibody in PBS supplemented with 2% BSA and 0.05% Triton X-100 (PBST) overnight at 4 °C, washed four times with PBST, and then incubated for 2 h with secondary antibody at RT. Glass coverslips were mounted with Vectashield mounting media. Primary antibodies were mouse anti-ABCA1 antibody (1:500, Abcam, ab18180) and rabbit anti-GFAP (1:500, Cell Signaling Technology, 12389 T). Secondary antibodies were FITC-conjugated goat anti-mouse IgG (1:500, Invitrogen, 31569) and AlexaFluor-594-conjugated donkey anti-rabbit IgG (1:500, Invitrogen, AB150076).

Confocal images of cultured mouse astrocytes were taken with an SP5 confocal laser-scanning microscope (Leica Microsystems). High-resolution optical sections (1024 × 1024 pixel format) were captured with a 20× zoom at 0.5 µm step intervals. For ABCA1 analysis, at least 10 images were captured per culture (100-200 astrocytes) with 4–5 cultures per group. In Image J, each z stack was collapsed into a single image by projection, and split by color. GFAP-expressing cells were outlined and saved in the ROI manager and used to measure the area and mean intensity of ABCA1 immunoreactivity in GFAP-expressing cells and corrected by subtracting the background intensity. Average intensity of ABCA1 immunoreactivity was calculated for each image (10–20 cells).

**Astrocyte conditioned medium.** For the collection of human ACM cells were plated on Matrigel-coated T25 flasks at a density of 20,000 cells/cm² and grown for 7 days. On the day prior to the sample collection, the medium was replaced with 5 ml of fresh neurosphere medium. As indicated cells were treated with all-trans retinoic acid (ATRA) for 24 h. Collected medium was filtered through a 0.22 µm filter and stored at −80 °C until use. Mouse ACM was collected one day after changing the medium to fresh DMEM supplemented with 10% FBS and 1× P-S. The samples were stored at −80 °C until use.

**Cholesterol assay.** The concentration of cholesterol was determined in ACM of human FXS and control astrocytes using the cholesterol Quantification kit (Sigma-Aldrich) and fluorometric detection according to the manufacturer´s instructions. All samples and standards were run in duplicate.

**Cytokine array.** Cytokine profile in ACM was analyzed using Human Cytokine Antibody Array (Abcam, Cam-bridge, UK, Ab133998). ACM was prepared by plating the cells on Matrigel-coated T25 flasks at a density of 20,000 cells per cm² and grown for 7 days. The medium was replaced with 5 mL of fresh NS a day prior to collecting the samples, which were filtered through a 0.22 µm filter and stored at −80 °C. For the analysis, media from four iPSC-derived astrocyte lines were pooled and array performed with undiluted medium according to the manufacturer's instructions. Chemiluminescent detection was performed with a ChemiDoc ima-ging system (Bio-Rad Finland, Helsinki, Finland), and densitometric data were obtained using ImageJ software. Mean intensities of negative and positive control spots were used for background correction and normalization, respectively.

**Liquid chromatography-mass spectrometry untargeted lipidomics.** Liquid chromatography-mass spectrometry (LC-MS) lipidomics analysis was performed at the UC Riverside Metabolomics Core Facility as described previously[78]. Briefly, analysis was performed on a Waters G2-XS quadrupole time-of-flight mass spectrometer coupled to a Waters Acquity I-class UPLC system. Separations were carried out on a Waters CSH C18 column (2.1 × 100 mm, 1.7 µM). The mobile phases were (A) 60:40 acetonitrile: water with 10 mM ammonium formate and 0.1% formic acid and (B) 90:10 isopropanol:acetonitrile with 10 mM ammonium formate and 0.1% formic acid. The flow rate was 400 µl/min and the column was held at 65 °C. The injection volume was 2 µl. The gradient was as follows: 0 min, 10% B; 1 min, 10% B; 3 min, 20% B; 5 min, 40% B; 16 min, 80% B; 18 min, 99% B; 20 min 99% B; 20.5 min, 10% B. The MS scan range was (50–1600 m/z) with a 100 ms scan time. MS/MS was acquired in a data-dependent fashion. Source and desolvation temperatures were 150 °C and 600 °C, respectively. Desolvation gas was set to 1100 l/h and cone gas to 150 l/h. All gases were nitrogen except the collision gas, which was argon. The capillary voltage was 1 kV in positive ion mode and 2 kV in negative mode. A quality control sample, generated by pooling equal aliquots of each sample, was analyzed periodically to monitor system stability and performance. 6 µl of injection volume was used for the data acquisition. Samples were analyzed in random order. Leucine enkephalin was infused and used for mass correction.

Untargeted data processing (peak picking, alignment, deconvolution, integration, normalization, and spectral matching) was performed in Progenesis Qi software (Nonlinear Dynamics). Data were normalized to total ion abundance of sample, total PL abundance, or total abundance of a specific lipid class. Features with a CV >30% across QC injections were removed[79,80]. To aid in the identification of features that belong to the same metabolite, features were assigned a cluster-ID using RAMClust[79,81]. An extension of the metabolomics standard initiative guidelines was used to assign annotation level confidence[82,83]. Lipids were identified according to the match of retention time, accurate MS m/z values, and MS/MS spectra compared to an in-house database generated with authentic standards (e.g., for desmosterol), the Lipidblast in-silico database[84] (e.g., for different PL species) or by using external databases including several mass spectral metabolite libraries in Mass Bank of North America, Metlin[85] (e.g., for cholesterol, and CE species detected and identified separately and summed up to CE total).

**RNA extraction and RT-PCR.** Total cellular RNA was extracted using NucleoSpin RNA kit (QIAGEN) according to the manufacturer's instructions and reverse-transcribed into complementary cDNA using iScriptTM cDNA synthesis kit (Bio-Rad, #170-8891). Using cDNA as template, RT-qPCR was performed with HOT FIREPol® EvaGreen® qPCR Mix Plus (Solis BioDyne) and LightCycler® 480 (Roche) for 45 cycles of 95 °C for 15 s, 62 °C for 20 s, and 72 °C for 20 s. Gene expressions were analyzed with the ΔΔCt method using GAPDH expression for normalization. Experiments were performed with three technical replicates. Astrocytes were treated with IL-1β (10 ng/ml) and TNF-α (50 ng/ml) for 7 days as indicated. Primers are shown in Table 1.

**RNA sequencing analysis.** RNA sequencing (RNA-seq) data were generated using triplicate samples of isogenic hPSC (H1 control and *FMR1* KO cell lines)-derived cells at 95 DIV and NextSeq500 platform. The sequencing reads were filtered based on quality and length, and after adapter removal the reads were aligned to the human genome using Star Aligner (version 2.5.0b) and counted with high-throughput sequencing. Differential expression analysis was performed with the R package DESeq2. Comparison of the three D95 FXS samples to the two isogenic control samples was performed to detect genes whose expression significantly differed.

**Statistics and reproducibility.** Data are presented as mean ± SEM. Student's *t* test was used to determine differences between control and FXS/*Fmr1* KO astrocytes, except for the expression of *CCL5* and *GFAP* in response to cytokines, where ANOVA was performed using SPSS Statistics 25 (IBM, Armonk, NY USA). Lipid variables in the experimental groups were demonstrated by heat maps (using Euclidian distances and showing a maximum of 50 most significantly differing variables by *t* test), Volcano plots (using fold change threshold 1.3, *P* value threshold 0.05, and false discovery rate correction) and principal component analysis 2D scores plot with 95% confidence regions or biplots (scores and

loadings). The analyses were performed by MetaboAnalyst 5.0 (freeware of Xia Lab @ McGill). Soft independent modeling of class analogies (SIMCA, a quantitative test using PCA-based group models)[86] was used to test whether the differences between the phospholipid species compositions of the WT and *Fmr1* KO sample groups were statistically significantly different. SIMCA was performed by Sirius V8.5 software (Pattern Recognition Systems PRS, Bergen, Norway). *P* value of <0.05 was considered statistically significant.

The experiments utilizing human astrocytes were performed with cells derived from at least three different FXS human iPSC lines representing biological patient-specific replicates and controls, except studies of cytokine responses where three separate cultures of both HEL11.4 control and HEL69.5 cell lines were used. The chemokine and cytokine profile of FXS and control astrocytes was shown in pooled samples. We used cultures derived from three *Fmr1* KO and WT mice in LC-MS analysis and 4–5 cultures of both genetic groups for ABCA1 expression studies.

**Reporting summary**. Further information on research design is available in the Nature Portfolio Reporting Summary linked to this article.

## Data availability

All data supporting the findings of this study are available from the corresponding author upon reasonable request. The RNA-seq data are available in the Gene Expression Omnibus database under accession number GSE228378. Source data for graphs and charts can be found in the Supplementary Data.

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

## Acknowledgements

We thank Jari Koistinaho for P02 and P04 control human iPSC lines and Mahmoud Pouladi for isogenic *FMR1* KO and control H1 hESC lines. We also thank the initial contribution of Ulla-Kaisa Peteri to the research project and services provided by the Biomedicum Imaging Unit and the FIMM Digital Microscopy, DNA Sequencing and Genomics Laboratory at the Institute of Biotechnology, and Molecular Pathology Unit supported by Helsinki University and Biocenter Finland, and Metabolomics Core supported by UCR. The work was supported by grants from FRAXA Research Foundation (to M.L.C. and I.M.E), the Academy of Finland, the Arvo and Lea Ylppö Foundation, and the Finnish Foundation for Pediatric Research. V.A.W. is supported by a TRANSCEND fellowship from the California Institute of Regenerative Medicine (Award # EDUC4-12752). Open access funded by Helsinki University Library.

## Author contributions

M.L.C. and I.M.E. conceived the project. R.K. provided additional input and analyzed lipid data. V.A.W., A.O.K., R.M., J.U., I.M.E., and M.L.C. performed experiments and analyzed data. J.S.K., A.B., and M.H. performed mass spectrometry analyses. K.T. analyzed data and contributed to the illustrations and finalization of the manuscript. M.L.C. and R.K. wrote the manuscript. All authors discussed the results and contributed to the manuscript.

## Competing interests

The authors declare no competing interests.
