## [Peer Review File · Communications Biology]

Reviewers' comments:

Reviewer #1 (Remarks to the Author):

The manuscript by Talvio et al. describes lipid dysregulations in human hiPSC-derived astrocytes derived from fragile X patients as well as in mouse Fmr1 knockout astrocytes versus controls. The authors found a defect in ABCA1 expression and subsequently a defect in cholesterol esterification, cholesterol synthesis, and also unsaturated fatty acid lipids. The scope of this manuscript is of descriptive nature, so that potentially interesting mechanistic experiments following those observations are not presented. However, within this scope, the manuscript presents original, new observations that may provide the basis for novel mechanistic insight into fragile X.

My comments concern mostly the presentation of data, it was not always clear to me with which method data was generated. Also, biological replicates are not mentioned.

1-Figure 1a: appears to be plus/minus SEM; figure legend indicates that only plus SEM is indicated.

2-Figure 1b: Scale bar is not indicated

3-Figure 1c: It is unclear how such quantifications were generated: Is this the quantification from ICC or from western blots? If they stem from ICC, the method of quantification is not described in the manuscript and needs to be added. If data stem from western blot, please add the western blot. Maybe add "ABCA1" as label on the x-axis.

4-The figure legend 1 indicates, that results stem from 3-4 control and 3-4 FXS iPSC lines, which is good. But it should be explicitly communicated in the figure legend if the experiments were repeated, i.e., were there 3 biological replicates of one experiment performed with 3-4 cell lines (or was each experiment done once with 3-4 cell lines)? In this sense, ideally the bar graphs should be replaced with scatter bar graphs.

5-Figure 1e/f: I believe the labeling of the bars needs to be revised, what is the shaded bar? Shading may resemble fig 1d? I assume "medium" means medium that has not been incubated on cells?

6-Figure 1g: is this ratio under conditions without RA?

7-Figure 2a: Is there a reason that ABCA1 expression by qPCR is depicted as "relative expression" in Fig 1a, but as "% of control" in fig 2a? Hard to compare this way.

8-Figure 2b/c: From the methods section, mouse astrocytes required 10% FBS. FBS itself contains lipids. For the cholesterol quantification from mouse astrocyte conditioned medium, was the FBS removed? Was the neurosphere medium used for quantifying cholesterol from mACM? If so, maybe add in the methods section that these samples underwent cholesterol quantification to make it clear to the reader.

9-Why would desmosterol be in the conditioned medium? Could the authors maybe comment on the abundance of desmosterol in medium?

10-Figure 2e: I assume this was a MS analysis? I would add that info to the figure legend.

11-Figure 2f: I would think that only a t-test is not correct in this analysis. Can the bar graphs be replaced by scatter bar graphs? Can the authors add an example mass spectrum to the supplemental data to show how the peaks look in mass spec for a significantly different PC species?

12-Clearly state if biological replicates have been obtained from the different cell lines.

13-In the paragraph introducing figure 3, I assume that this is now mass spec lipidomic analysis? Please add the method to either text or figure legend.

14-Why is the cholesterol content in figure 2 not significantly different, but in figure 3 it is?

15-Figure 3e: as I understood from the methods, only a t-test was performed, I believe the statistical testing needs revision

16-Figure 4c/d: the used method is neither mentioned in text nor figure legend, was this mRNA expression by qPCR, ELISA, western blot? It would be good to expression levels of untreated cells

in comparison to treated cells.

17-Does IL13 treatment rescue ABCA1 expression in FXS astrocytes?

18-For figure 4 again, please be explicit about biological replicates of different cell lines.

19-There is no data provides for the last sentence of the results section referring to RNAseq data and TNRF5F10D. I suggest deleting this sentence.

Reviewer #2 (Remarks to the Author):

The paper entitled "Dysregulated cholesterol homeostasis in astrocytes modeling fragile X syndrome" by Talvio and Colleagues analyzes ABCA1 in particular, and cholesterol homeostasis more in general, in astrocytes generated from human induced pluripotent stem cells (hiPSCs) derived from males with fragile X syndrome (FXS) but also on other experimental models. Their results demonstrate that changes of lipid metabolism occur in astrocytes; These changes can affect membrane properties and cholesterol transport in FXS astrocytes, providing target for therapy in this pathology.

The paper is interesting, well written, the statistical analysis carried on correctly. Moreover, it adds new insights in this pioneering field.

This reviewer only suggests to make the graphs more readable. In fact, while in figure 1 any graph shows a title or the correct description on Y axe, in panels a, b, d and even f of figure 2, the reader must read the figure legend to understand. Usually, the figures in scientific papers should be self-explanatory.

Reviewer #3 (Remarks to the Author):

Interesting paper with novel results about the lipids involvement in the FXS pathophysiology. The present study shows that FMRP-deficiency in astrocytes is associated with a reduce ABCA1 expression in both human and mice models as well as a dys-balance of the membrane lipids composition.

Few suggestions:

Please reformulate the second sentence of the first paragraph of the introduction part;

I suggest to include a formal reference instead of OMIM:300624;

The authors often show results of reduced levels of ABCA1. Please specify ABCA1 m-RNA expression when appropriate in order to facilitate the comprehension of the results.

Figure 2 (a,b,d): the text of the y axis might be more explicit. Instead of the "% of control" the authors might write "relative ABC1 immunoreactivity", "cholesterol relative abundance" and "desmosterol relative abundance".

It is important to specify that astrocytes are mainly involved in the regulation of the developed brain (second sentence of the discussion).

The authors mention briefly the potential benefits of omega-3 supplementation. I suggest authors might discuss more about that, including data from a recent paper of Abolghasemi et al 2022 about the alteration of fatty acid profile in FXS plasma humans;

Helsinki, 3.1.2023

Response to the Reviewer Comments

Reviewer #1.

We thank the Reviewer for thoughtful assessments of the manuscript. The manuscript has been revised to address concerns as follow:

1-Figure 1a: appears to be plus/minus SEM; figure legend indicates that only plus SEM is indicated.

Response: The figure legends are revised to \pm SEM.

2-Figure 1b: Scale bar is not indicated

Response: Scale bar of panel 1b is defined (50 μ m) in the figure legend of the revised manuscript.

3-Figure 1c: It is unclear how such quantifications were generated: Is this the quantification from ICC or from western blots? If they stem from ICC, the method of quantification is not described in the manuscript and needs to be added. If data stem from western blot, please add the western blot. Maybe add “ABCA1” as label on the x-axis.

Response: ICC was used to obtain the data shown in panel 1c. The method is described and x-axis is revised to “relative ABCA1 immunoreactivity (%)”.

4-The figure legend 1 indicates, that results stem from 3-4 control and 3-4 FXS iPSC lines, which is good. But it should be explicitly communicated in the figure legend if the experiments were repeated, i.e., were there 3 biological replicates of one experiment performed with 3-4 cell lines (or was each experiment done once with 3-4 cell lines?)? In this sense, ideally the bar graphs should be replaced with scatter bar graphs.

Response: We have replaced the bar graphs with scatter bar graphs in Figure 1 and revised the figure legend to clearly indicate that the results were always obtained from experiments with 3-4 biological replicates (n = cell lines). RT-PCR experiments were performed with three technical replicates (n), which is added to Methods. The number of analyzed immunoreactive cells [$n(\text{control}) = 214$ and $n(\text{FXS}) = 428$ cells on 3-4 coverslips/each cell line of both genotypes] has been added to Figure legend.

5-Figure 1e/f: I believe the labeling of the bars needs to be revised, what is the shaded bar? Shading may resemble fig 1d? I assume “medium” means medium that has not been incubated on cells?

Response: The shaded bar indicated medium that was not incubated with cells and showed levels of cholesterol and cholesteryl ester. To clarify, the shaded bars have been replaced by dashed lines in the revised manuscript. Figure legend has been revised to “*Dashed line shows cholesterol (e) and cholesteryl ester (f) levels in medium that was not incubated with cells.*”

6-Figure 1g: is this ratio under conditions without RA?

Response: The ratio was increased under both untreated and treated conditions. Data shown in panel 1g were obtained under basal conditions without RA treatment and this information has been added to Figure legend.

7-Figure 2a: Is there a reason that ABCA1 expression by qPCR is depicted as “relative expression” in Fig 1a, but as “% of control” in fig 2a? Hard to compare this way.

Response: Figure 2a shows reduced ABCA1 protein immunoreactivity in FXS mouse astrocytes as “% of control” and the data correlate with the data of ABCA1 immunoreactivity in human FXS astrocytes shown in Figure 1c similarly as “% of control”, whereas in Figure 1a relative expression of mRNA obtained by RT-PCR analysis is shown. We have revised labelling of the figures to clarify the data.

8-Figure 2b/c: From the methods section, mouse astrocytes required 10% FBS. FBS itself contains lipids. For the cholesterol quantification from mouse astrocyte conditioned medium, was the FBS removed? Was the neurosphere medium used for quantifying cholesterol from mACM? If so, maybe add in the methods section that these samples underwent cholesterol quantification to make it clear to the reader.

Response: Mouse astrocytes were cultured in DMEM supplemented with 10% FBS and 1x penicillin-streptomycin. Since mACM contained FBS, we have revised Methods to provide this important information to the readers “*Mouse ACM was collected one day after changing the medium to fresh DMEM supplemented with 10% FBS and 1 x P-S.*”

9-Why would desmosterol be in the conditioned medium? Could the authors maybe comment on the abundance of desmosterol in medium?

Response: We did not observe any significant change in free cholesterol in human and mouse astrocyte medium, but we found increased cholesterol content in *FMRI* KO astrocytes, indicating increased synthesis instead of uptake. In addition, our data show a decreased desmosterol levels in mACM with an increase in mouse KO astrocytes suggesting higher uptake of desmosterol, the precursor that is needed to synthesize more cholesterol, from the medium of *Fmr1* KO astrocytes containing serum. The discussion was added to the manuscript.

10-Figure 2e: I assume this was a MS analysis? I would add that info to the figure legend.

Response: We have added “*LC-MS analysis*” to the figure legend of Figure 2e.

11-Figure 2f: I would think that only a t-test is not correct in this analysis. Can the bar graphs be replaced by scatter bar graphs? Can the authors add an example mass spectrum to the supplemental data to show how the peaks look in mass spec for a significantly different PC species?

Response: We performed nonparametric statistical analysis (SIMCA, a quantitative test using PCA based group models) to confirm the significant differences obtained in *t*-test. The statistical confirmation has been added to Figure legend and Methods has been revised accordingly. A new Supplementary Figure 1 shows LC-MS ion chromatograms as requested by the reviewer.

12-Clearly state if biological replicates have been obtained from the different cell lines.

Response: The experiments were performed with three biological replicates. In addition, in lipidomic analysis FXS data (n = 4) included two FXS clones derived from one FXS donor.

Cell lines used in the studies are provided in Methods and we have added the information about the cell lines representing biological variants in Figure legends.

13-In the paragraph introducing figure 3, I assume that this is now mass spec lipidomic analysis? Please add the method to either text or figure legend.

Response: Mass spectrometry was used to obtain data shown in Figure 3 and the method is defined in the figure legend.

14-Why is the cholesterol content in figure 2 not significantly different, but in figure 3 it is?

Response: The cholesterol content was not affected in ACM of human (Figure 1) or mouse FXS astrocytes (Figure 2) with abnormally reduced ABCA1 expression, suggesting activation of mechanisms that maintain extracellular cholesterol level normal. In Figure 3 the increase of cholesterol in FXS mouse astrocytes is shown, which can be a result of accumulation of cellular cholesterol in agreement with reduced ABCA1 expression and increased cholesterol synthesis supported by increased desmosterol uptake. These differences between cellular cholesterol content in FXS and control mouse astrocytes associated with maintenance of normal extracellular cholesterol are now discussed in the revised manuscript.

15-Figure 3e: as I understood from the methods, only a t-test was performed, I believe the statistical testing needs revision

Response: Soft independent modeling of class analogies (SIMCA, a quantitative test using PCA based group models) was used to confirm significant differences between the phospholipid species compositions of the WT and *Fmr1* KO sample groups. The statistics is added to Figure legend and described in Methods.

16-Figure 4c/d: the used method is neither mentioned in text nor figure legend, was this mRNA expression by qPCR, ELISA, western blot? It would be good to expression levels of untreated cells in comparison to treated cells.

Response: The data shown in Figure 4c/d were obtained using RT-qPCR using the $\Delta\Delta C_t$ method for analysis. Relative expression of *CCL5* mRNA to *GAPDH* expression is shown in FXS and control astrocytes, which are previously characterized to express similarly astrocyte marker GFAP (Peteri et al. *GLIA*, 2021). Treatment effects of IL-1 β and TNF- α on *CCL5* and *GFAP* mRNA expression were analyzed relative to controls by RT-qPCR using the $\Delta\Delta C_t$ method. The method used is described in Figure legend.

17-Does IL13 treatment rescue ABCA1 expression in FXS astrocytes?

Response: Our present studies did not investigate effects of IL13 treatment. The potential rescue effects of IL13 treatment on ABCA1 expression were discussed based on previous studies of Ma et al. and the appropriate reference has been added to the revised text in Discussion.

18-For figure 4 again, please be explicit about biological replicates of different cell lines.

Response: We provide the number of cell lines (n) as the number of biological replicates.

19-There is no data provides for the last sentence of the results section referring to RNAseq data and TNFRSF10D. I suggest deleting this sentence.

Response: The sentence has been omitted and the reference list has been revised accordingly.

Reviewer #2

We thank the Reviewer for careful assessment of the manuscript. The manuscript has been revised to address each concern as follow:

This reviewer only suggests to make the graphs more readable. In fact, while in figure 1 any graph shows a title or the correct description on Y axe, in panels a, b, d and even f of figure 2, the reader must read the figure legend to understand. Usually, the figures in scientific papers should be self-explanatory.

Response: Figures have been extensively revised to make them self-explanatory.

Reviewer #3

We thank the Reviewer for thoughtful assessment of the manuscript. The manuscript has been revised to address each concern as follow:

Please reformulate the second sentence of the first paragraph of the introduction part.

Response: The sentence is revised to *“It is crucial for synapse formation and function as high cholesterol content is required in lipid rafts.”*

I suggest to include a formal reference instead of OMIM:300624;

Response: Text has been revised with a formal reference.

The authors often show results of reduced levels of ABCA1. Please specify ABCA1 m-RNA expression when appropriate in order to facilitate the comprehension of the results.

Response: Figures and figure legends have been revised to clarify the results.

Figure 2 (a,b,d): the text of the y axis might be more explicit. Instead of the “% of control” the authors might write “relative ABC1 immunoreactivity”, “cholesterol relative abundance” and “desmosterol relative abundance”.

Response: Figures and figure legends have been revised as suggested.

It is important to specify that astrocytes are mainly involved in the regulation of the developed brain (second sentence of the discussion).

Response: Discussion has been revised.

The authors mention briefly the potential benefits of omega-3 supplementation. I suggest authors might discuss more about that, including data from a recent paper of Abolghasemi et al 2022 about the alteration of fatty acid profile in FXS plasma humans;

Response: The figures and figure legends have been revised to clarify the results.

REVIEWERS' COMMENTS:

Reviewer #1 (Remarks to the Author):

The authors improved the description of the experiments, figures and figure legends significantly. I believe the manuscript and experiments are now clearly explained and were conducted with high experimental standards in sufficient replicates.

Helsinki, 30.3.2023

Communications Biology

To the Reviewers,

We have revised the manuscript retitled according to the editor's suggestion "*An iPSC-derived astrocyte model of fragile X syndrome exhibits dysregulated cholesterol homeostasis*" as requested by the editorial policy of *Nature Journals*. These changes include Abstract in present tense, changes in Figure legends, all data presentation in graphs with individual data points, a separate section titled "Statistics and Reproducibility" in Methods, GEO submission number of RNA-seq data, exact p-values, and a new format of the Supplementary Figure 1.

When summarizing the data used in the figures to an excel sheet we found that the lipids data of *Fmr1* KO and wild type mouse astrocytes were handled the wrong way around, which we have corrected in the revised version. The cholesterol data did not change. The most important change was the increase of polyunsaturated lipids in *Fmr1* KO astrocytes compared with controls, which requested us to revise Discussion and Figure 5 slightly. These changes are highlighted in yellow in the text.

We feel that the final revised manuscript provide solid data to be published in *Communcations Biology*.

Sincerely,

Maija Castrén, MD, PhD, Child Neurologist

Reviewers' comments:

Reviewer #2 (Remarks to the Author):

I have read the revised version of the paper, checked the changes Authors have done as well as the changes in the presented results.

I do not have any concerns.

Reviewer #4 (Remarks to the Author):

Summary of the outstanding features of the work:

Defects in the enzymes of cholesterol synthesis pathway result in developmental disorders and life-long consequences. Altering cholesterol levels in brains during development, either by gene dysregulation or pharmaceutical treatment, may impair brain functions including cellular signaling as well as synaptic development and function.

Decreased cholesterol has been previously reported in serum and platelets of individuals with FXS (1-3) as well as in the serum of an FXS rat model (4). Cholesterol measurement in the brain of the rat FXS model showed it was increased in a single brain region though the study used an approach that cannot differentiate between cholesterol precursors.

Multiple signaling pathways were found dysregulated in the Fmr1 KO mouse including the Ras-ERK 1/2 signaling pathway downstream of mGluRs, which results in enhanced protein synthesis. This pathway is targeted in FXS preclinical studies (5-7) and clinical trials (8-10) by administering lovastatin which can reduce Ras-ERK 1/2 signaling.

The authors used iPSCs derived from males with Fragile X syndrome and differentiated them into astrocytes. This is highly relevant and important biological material used in this study. Cholesterol biosynthesis (the whole pathway includes over 30 enzymes) and metabolism are complex. The authors analyzed cholesterol levels in the medium but did not study cholesterol synthesis within the cells. It is difficult to make final conclusions about cholesterol metabolism in FXS without considering both cellular cholesterol synthesis, and efflux from the cells.

Only four experimental and five control cell lines were used in the study. Considering high variability in humans, some of described data may not reproduce in much larger study using many more samples. The cholesterol measurements in cell culture medium were done using a kit and mass spec. The kit measures cholesterol and cannot differentiate among cholesterol, desmosterol and other sterols. Mass spec is a great method to measure cholesterol but from description it is not clear if the authors used cholesterol and desmosterol standards to verify the identity of chromatography peaks. In addition to cholesterol and desmosterol, there are other sterols that may affect the final cholesterol values (including 7-dehydrocholesterol, zymosterol, zymostenol, lathosterol, dehydrolathosterol, dehydrodesmosterol and others). The mouse astrocytes were grown in medium with 10% FBS and the sterols were measured in medium with high level of cholesterol. If cholesterol is studied in cultures and if the cells synthesize their own cholesterol, the cells need to be grown in cholesterol-deficient medium. Additional experiments feeding cells either labelled acetate or labeled glucose and measuring labeled cholesterol would be conclusive about sterol biosynthesis and efflux. RNA sequencing data: the authors present sequencing data for ABCA1 and SLC27A1 mRNA expression and refer the source of data to reference 25. In reference #25 I could not find source data for ABCA1 and SLC27A1; these two mRNAs are not listed as changed in reference #25. Primer sequences: some sequences are mouse, other humans. Additional details should be included showing the accession number of the sequence that was used to design these primers. If both human and mouse primers were used, all should be included. Detailed phospholipid changes in the membranes are interesting data but without growing cells in cholesterol deficient medium, it is difficult to make conclusions about cholesterol accumulation within cells, phospholipid membrane changes and cytokine expression.

Additional comments:

On page 3 authors state: "Treatment with lovastatin, an inhibitor of Hmgcr which regulates the early irreversible and rate-limiting step in the biosynthesis of cholesterol, dampens neuronal

hyperexcitability in the brain of FSX mouse model, Fmr1 KO mice, and rescues part of the mouse FXS phenotype (14)“ The authors are correct about lovastatin inhibiting Hmgcr. However, the Osterweil et al did not measure levels of cholesterol in the brain. Osterweil emphasized in the manuscript that lovastatin is used as inhibitor of protein synthesis and not to lower cholesterol. Lovastatin decreased protein synthesis and prevented epileptogenesis.

On page 3 authors say: “Beneficial effects of lovastatin treatment have also been observed on behavior of individuals with FXS in several clinical trials (15-17). The authors should provide specific details what were the beneficial effects and if lovastatin is indeed current treatment for FXS.

In RESULTS, first paragraph: Reduced ABCA1 expression in human FXS astrocytes, there is sentence: “Similarly, human embryonic stem cell (hESC)-derived FMR1 KO astrocytes expressed less ABCA1 than their isogenic controls in RNA Seq analysis (Log2 -6.41-fold change, P=0.0004, P adjust = 0.125)25. I read carefully published reference 25 and could not find any data about ABCA1 downregulation.

Figure 1B. ABCA1 immunofluorescence shows as nuclear stain; if ABCA1 is transporter, the staining should be membrane staining. There are cells that express ABCA1 but are not GFAP positive. ABCA1 Ab on Abcam website is not convincing image for IF.

Figure 1D. Very confusing figure without sufficient description. The numbers below graph represent different cell lines used in a study – what does HEL mean? Does each bar represent one technical replicate? Conclusion: “total cholesterol content did not differ between FXS and control astrocyte conditioned medium”. “The concentration of cholesterol was determined in ACM of human FXS and control astrocytes using the cholesterol Quantification kit (Sigma-Aldrich) and fluorometric detection according to the manufacturer’s instructions”. This is not reliable method for cholesterol quantification. Especially when medium used in the experiment contains 10% FBS (mouse ACM was DMEM plus 10% FBS). In the excel supplementary info, the authors show experimental details for cholesterol fluorometric analysis and the numbers are % of control. What is % of control for the control cells?

Figure 1E. Cholesterol level relative abundance measured by mass spec: description of mass spec cholesterol measurement is missing. Why are numbers 0 to 14,000? What does the dashed line with italic medium represent? Again, no difference in cholesterol level detected in ACM by mass spec between control and FXS.

Figure 1F. There is no information how is cholesteryl ester measured. There is no clear description of the Figure. Was alkyne hydrolysis used to remove esters? What does dotted line labeled with Medium shows?

Figure 1H. In results section authors wrote: “Similar to ABCA1 expression, SLC27A1 was reduced in FXS iPSC-derived astrocytes (Fig 1h) and seen in hESC-derived astrocytes lacking FMRP (log2 - 7.38-fold change, P=0.034) 25 when compared with controls”. I read carefully published reference 25 and could not find any data about SLC27A1 or FATP1 downregulation.

The golden standard units for presenting qPCR data and relative mRNA expression are delta delta Ct. The authors use relative mRNA expression and have units from 0 to 0.2 and 0 to 0.001. Based on the presentation it is impossible to understand the level of expression of ABCA1 and FAT1P in astrocytes. Is GAPDH the best normalizer? The ideal normalizer should be at the expression level similar to the actual gene of interest. It seems that GAPDH is expressed at very high levels and the two genes of interest are present at extremely low levels.

The authors should provide the accession number for sequences that were used to generate primers. Were the mouse primers used to amplify human genes? The primer sequences for ABCA1 and GAPDH published in Table 1 show primers that are specific to mice; not human.

Figure 2a. ABCA1 relative immunoreactivity: difference is really small and there is lack of description how is this measurement done. In the Supplementary Excel file there is list of images and WT % of control and Fmr1 KO % of control. What are controls for WT and what are controls for Fmr1 KO cells? If it is %, the graph in 2a shows scale 0-250% and the numbers in excel show 0.5 – 2.0.

Figure 2b,c,d show relative abundance of cholesterol, cholesteryl ester and desmosterol in ACM. This is very problematic measurement because the mouse astrocytes were grown in DMEM with 10% FBS. Was the cellular content measured? What is the baseline cholesterol level in the medium?

The conclusion on page 7 top paragraph: “The altered PC profile of Fmr1 KO ACM suggested that the absence of FMRP led to dysregulation of ABCA1-mediated efflux of PC along with cholesterol from Fmr1 KO astrocytes.” The data presented do not support this conclusion. The measurement

of ABCA1 immunoreactivity is not the best method and the changes presented are very small. Western blotting would be better choice compared to measuring fluorescence intensity on the microscope. While PC changes look convincing, the measurement of cholesterol is not ideal. Most of cholesterol measurement was done in medium containing 10% FBS and using a kit (and not mass spec). Even when mass spec was used, the authors did not use established standards to verify the identity of sterol peaks. The proper experiment would be feeding cells labeled precursors and precise quantification of labeled products in the medium. These experiments were not done.

Figure 5. Cholesterol biosynthesis starts with two molecules of acetyl CoA and formation of acetoacetyl-CoA. This is followed by a second condensation of acetyl CoA and acetoacetyl-CoA to form 3-hydroxy-3-methylglutaryl CoA (HMG-CoA) which is shown in the figure. The figure shows cholesterol precursors – these should be acetylCoA. Cholesterol immediate precursors are desmosterol and 7-dehydrocholesterol. The schematic shows elevated cholesterol in the cell, but the authors analyzed cholesterol in the medium. Desmosterol was found decreased in the mouse FXS astrocytes condition medium and elevated in cells. Desmosterol was not analyzed in human astrocytes. While it is important to summarize data this manuscript does not show the mechanisms involved in cholesterol homeostasis.

References: marked in yellow are references that author used.

1. Berry-Kravis E, Levin R, Shah H, Mathur S, Darnell JC, Ouyang B. Cholesterol levels in fragile X syndrome. *Am J Med Genet A*. 2015 Feb;167A(2):379-84. doi: 10.1002/ajmg.a.36850. Epub 2014 Nov 25. PMID: 25424470; PMCID: PMC5436717. During the course of follow up of a large cohort of patients with FXS we noted that many patients had low cholesterol and HDL values and thus initiated a systematic chart review of all cholesterol values present in charts from a clinic cohort of over 500 patients with FXS. Total cholesterol (TC), low density lipoprotein (LDL) and high density lipoprotein (HDL) were all significantly reduced in males from the FXS cohort relative to age-adjusted population normative data.
 2. Çaku A, Seidah NG, Lortie A, Gagné N, Perron P, Dubé J, Corbin F. New insights of altered lipid profile in Fragile X Syndrome. *PLoS One*. 2017 Mar 23;12(3):e0174301. doi: 10.1371/journal.pone.0174301. PMID: 28334053; PMCID: PMC5363930.
 3. Lisik MZ, Gutmajster E, Sieroń AL. Low Levels of HDL in Fragile X Syndrome Patients. *Lipids*. 2016 Feb;51(2):189-92. doi: 10.1007/s11745-015-4109-6. Epub 2015 Dec 28. PMID: 26712713; PMCID: PMC4735238.
 4. Parente, M.; Tonini, C.; Buzzelli, V.; Carbone, E.; Trezza, V.; Pallottini, V. Brain Cholesterol Biosynthetic Pathway Is Altered in a Preclinical Model of Fragile X Syndrome. *Int. J. Mol. Sci.* 2022, 23, 3408. <https://doi.org/10.3390/ijms23063408>
- Note: cholesterol spectrometric assay - no mass spectrometry
5. Osterweil EK, Chuang SC, Chubykin AA, Sidorov M, Bianchi R, Wong RK, Bear MF. Lovastatin corrects excess protein synthesis and prevents epileptogenesis in a mouse model of fragile X syndrome. *Neuron*. 2013 Jan 23;77(2):243-50. doi: 10.1016/j.neuron.2012.01.034. PMID: 23352161; PMCID: PMC3597444. (no measurements of cholesterol; lovastatin used as inhibitor of protein synthesis; not to lower cholesterol)
 6. Pellerin D, Çaku A, Fradet M, Bouvier P, Dubé J, Corbin F. Lovastatin corrects ERK pathway hyperactivation in fragile X syndrome: potential of platelet's signaling cascades as new outcome measures in clinical trials. *Biomarkers*. 2016 Sep;21(6):497-508. doi: 10.3109/1354750X.2016.1160289. Epub 2016 Apr 8. PMID: 27058300. pERK and pAKT increased in FXS platelets and lovastatin normalized ERK activity – no measurement of cholesterol.
 7. Asiminas A, Jackson AD, Louros SR, Till SM, Spano T, Dando O, Bear MF, Chattarji S, Hardingham GE, Osterweil EK, Wyllie DJA, Wood ER, Kind PC. Sustained correction of associative learning deficits after brief, early treatment in a rat model of Fragile X Syndrome. *Sci Transl Med*. 2019 May 29;11(494):eaao0498. doi: 10.1126/scitranslmed.aao0498. PMID: 31142675; PMCID: PMC8162683. Note: Lovastatin in food – no measurements of cholesterol – just behavioral response.
 8. Çaku A, Pellerin D, Bouvier P, Riou E, Corbin F. Effect of lovastatin on behavior in children and adults with fragile X syndrome: an open-label study. *Am J Med Genet A*. 2014 Nov;164A(11):2834-42. doi: 10.1002/ajmg.a.36750. Epub 2014 Sep 24. PMID: 25258112. Safety study – no major side effect BUT also no real effect
 9. Thurman AJ, Potter LA, Kim K, Tassone F, Banasik A, Potter SN, Bullard L, Nguyen V, McDuffie A, Hagerman R, Abbeduto L. Controlled trial of lovastatin combined with an open-label treatment

of a parent-implemented language intervention in youth with fragile X syndrome. *J Neurodev Disord.* 2020 Apr 22;12(1):12. doi: 10.1186/s11689-020-09315-4. PMID: 32316911; PMCID: PMC7175541. Not much of LOV effect (same as controls)

10. Champigny C, Morin-Parent F, Bellehumeur-Lefebvre L, Çaku A, Lepage JF, Corbin F. Combining Lovastatin and Minocycline for the Treatment of Fragile X Syndrome: Results From the LovaMiX Clinical Trial. *Front Psychiatry.* 2022 Jan 4;12:762967. doi: 10.3389/fpsyt.2021.762967. PMID: 35058813; PMCID: PMC8763805.

Helsinki, 11.6.2023

Response to the reviewer comments

We thank the reviewers for careful assessments of the manuscript. The manuscript has been revised to address each concern as follow. The responses are highlighted in yellow in the revised manuscript.

Reviewer #2:

I have read the revised version of the paper, checked the changes Authors have done as well as the changes in the presented results. I do not have any concerns.

Reviewer #4:

Summary of the outstanding features of the work: Defects in the enzymes of cholesterol synthesis pathway result in developmental disorders and life-long consequences. Altering cholesterol levels in brains during development, either by gene dysregulation or pharmaceutical treatment, may impair brain functions including cellular signaling as well as synaptic development and function.

Decreased cholesterol has been previously reported in serum and platelets of individuals with FXS (1-3) as well as in the serum of an FXS rat model (4). Cholesterol measurement in the brain of the rat FXS model showed it was increased in a single brain region though the study used an approach that cannot differentiate between cholesterol precursors. Multiple signaling pathways were found dysregulated in the Fmr1 KO mouse including the Ras-ERK $\frac{1}{2}$ signaling pathway downstream of mGluRs, which results in enhanced protein synthesis. This pathway is targeted in FXS preclinical studies (5-7) and clinical trials (8-10) by administering lovastatin which can reduce Ras-ERK $\frac{1}{2}$ signaling.

The authors used iPSCs derived from males with Fragile X syndrome and differentiated them into astrocytes. This is highly relevant and important biological material used in this study. Cholesterol biosynthesis (the whole pathway includes over 30 enzymes) and metabolism are complex. The authors analyzed cholesterol levels in the medium but did not study cholesterol synthesis within the cells. It is difficult to make final conclusions about cholesterol metabolism in FXS without considering both cellular cholesterol synthesis, and efflux from the cells. Only four experimental and five control cell lines were used in the study. Considering high variability in humans, some of described data may not reproduce in much larger study using many more samples.

RESPONSE:

Our study shows reduced ABCA1 expression in both human and mouse FXS astrocytes. Altered ABCA1 expression did not associate with changes in total cholesterol content in ACM. However, the ratio of cholesterol/cholesterol ester was increased in human ACM due to an increase in non-ester cholesterol in FXS media.

Under current culture conditions, in defined medium without serum (Advanced DMEM/F12 culture medium), analysis of human iPSC-derived astrocyte pellets were not enough to yield

measurable results in analysis of lipids using LC-MS. We agree with the reviewer that different culture conditions or variability in human astrocyte genetic makeup most likely contributed to the results. However, addressing this was beyond the scope of current study and future study will delve into the effects of different media conditions on growth and lipid content of the human cells with genetically characterized human iPSC-derived FXS and control astrocytes. This was one of the reasons why we studied by LC-MS analysis the astrocytes derived from inbred Fmr1 KO mice, in both media and cell lysates.

The cholesterol measurements in cell culture medium were done using a kit and mass spec. The kit measures cholesterol and cannot differentiate among cholesterol, desmosterol and other sterols. Mass spec is a great method to measure cholesterol but from description it is not clear if the authors used cholesterol and desmosterol standards to verify the identity of chromatography peaks. In addition to cholesterol and desmosterol, there are other sterols that may affect the final cholesterol values (including 7-dehydrocholesterol, zymosterol, zymostenol, lathosterol, dehydrolathosterol, dehydrodesmosterol and others). The mouse astrocytes were grown in medium with 10% FBS and the sterols were measured in medium with high level of cholesterol. If cholesterol is studied in cultures and if the cells synthesize their own cholesterol, the cells need to be grown in cholesterol-deficient medium. Additional experiments feeding cells either labelled acetate or labeled glucose and measuring labeled cholesterol would be conclusive about sterol biosynthesis and efflux.

RESPONSE:

The cholesterol measurements were performed using both a kit and MS in human ACM collected from astrocytes cultured without serum in Advanced DMEM/F12, which allows serum reduction due to the added ethanolamine, glutathione, ascorbic acid, insulin, transferrin, AlbuMAX™ II lipid-rich bovine serum albumin for cell culture, and the trace elements sodium selenite, ammonium metavanadate, cupric sulfate, and manganous chloride. Lipids were identified according to the match of retention time, accurate MS m/z values and MS/MS spectra compared to an in-house database generated with authentic standards (e.g., for desmosterol), the Lipidblast in-silico database (e.g., for different PL species) or by using external databases including several mass spectral metabolite libraries in Mass Bank of North America, Metlin (e.g., for cholesterol, and CE species detected and identified separately and summed up to CE total). The MS data did not show significant differences between sterols of FXS and control human astrocytes, as shown in cholesterol and cholesterol ester amounts, but the cholesterol/cholesterol ester ratio was increased. Mouse astrocyte studies revealed changes of desmosterol content in Fmr1 KO astrocytes and conditioned medium, cultured with serum, that are consistent with an increase in desmosterol uptake by FXS mouse astrocytes, providing a substrate for cholesterol synthesis.

RNA sequencing data: the authors present sequencing data for ABCA1 and SLC27A1 mRNA expression and refer the source of data to reference 25. In reference #25 I could not find source data for ABCA1 and SLC27A1; these two mRNAs are not listed as changed in reference #25. Primer sequences: some sequences are mouse, other humans. Additional details should be included showing the accession number of the sequence that was used to design these primers. If both human and mouse primers were used, all should be included.

RESPONSE:

ABCA1 and SLC27A1 were found to be reduced in RNA Seq data of embryonic stem cell-derived human FXS astrocytes, which carry a mutation in the FMR1 gene produced by gene editing, compared with the isogenic control. The RNA Seq data are published in GLIA, reference #25, where only genes with p adjust < 0.1 were listed. The RNA Seq data are available at the Gene Expression Omnibus database under accession number GSE228378. In the present study, the reduced expression of ABCA1 and SLC27A1 were confirmed in iPSC-derived

astrocytes derived from 3 control and 4 different cell lines. The reference indicating the RNA Seq data used for ABCA1 expression analysis has been replaced in the revised manuscript. The reduced SLC27A1 expression in the RNA Seq data has been omitted for the avoidance of doubt and because the results were shown appropriately in biological replicates. Since all mRNA studies were performed in human samples, the primer sequences are human as indicated in the revised manuscript and the studies are based on Kielar et al. 2021, <https://academic.oup.com/clinchem/article/47/12/2089/5639397> and Alicea et al. 2020, <https://www.ncbi.nlm.nih.gov/pmc/articles/PMC7483379/>.

We have added the accession numbers of sequences used to design the primers: AF275948 (human ABCA1 mRNA); NM_002985 (human CCL5 mRNA); JO2642 (human GAPDH mRNA); NM_002055 (human GFAP mRNA); NM_198580 XM_32251 (human SLC27A1 mRNA).

Detailed phospholipid changes in the membranes are interesting data but without growing cells in cholesterol deficient medium, it is difficult to make conclusions about cholesterol accumulation within cells, phospholipid membrane changes and cytokine expression.

RESPONSE:

Mouse astrocytes cultured with serum represent well known experimental setting, allowing passaging of mouse astrocytes in culture. Under these conditions, using LC-MS we found that desmosterol was increased in cells but decreased in the medium, suggesting uptake of desmosterol. The cholesterol was also increased in cells but did not change in the medium. Although we completed the study with a detailed analysis of phospholipids in mouse astrocytes, we agree that the results may be not relevant for conclusions about cholesterol accumulation and we have removed the phospholipid details from the summary Figure 5. The cytokine expression was studied in human astrocytes under serum free condition in Advanced DMEM/F12 medium supplemented with ethanolamine, glutathione, ascorbic acid, insulin, transferrin, AlbuMAX™ II lipid-rich bovine serum albumin for cell culture, and the trace elements sodium selenite, ammonium metavanadate, cupric sulfate, and manganous chloride. To emphasize studies performed in human vs mouse astrocytes, we have color-coded human and mouse findings. Please note that ABCA1 decrease was seen in both types of astrocytes.

Additional comments:

On page 3 authors state: “Treatment with lovastatin, an inhibitor of Hmgcr which regulates the early irreversible and rate-limiting step in the biosynthesis of cholesterol, dampens neuronal hyperexcitability in the brain of FSX mouse model, Fmr1 KO mice, and rescues part of the mouse FXS phenotype (14)” The authors are correct about lovastatin inhibiting Hmgcr. However, the Osterweil et al did not measure levels of cholesterol in the brain. Osterweil emphasized in the manuscript that lovastatin is used as inhibitor of protein synthesis and not to lower cholesterol. Lovastatin decreased protein synthesis and prevented epileptogenesis.

On page 3 authors say: “Beneficial effects of lovastatin treatment have also been observed on behavior of individuals with FXS in several clinical trials (15-17). The authors should provide specific details what were the beneficial effects and if lovastatin is indeed current treatment for FXS.

RESPONSE:

The text on page 3 has been revised to better introduce the aim of our study to explore mechanisms conferring beneficial effects of lovastatin in FXS observed in the rodent FXS models and some pilot clinical studies. There is an evidence that platelets isolated from lovastatin-treated FXS patients display reduced ERK signaling that contributes to the excessive protein

synthesis in FXS (Pellerin et al., 2016, <https://pubmed.ncbi.nlm.nih.gov/27058300/>). Lovastatin may inhibit the Ras farnesylation required for membrane association and subsequent activation of ERK pathway (Mendola and Backer, 1990, <https://pubmed.ncbi.nlm.nih.gov/2278880/>), but differential effects of simvastatin compared to that of lovastatin question attenuation of farnesylation as the only reason for beneficial effects of lovastatin (Muscas et al., 2019, <https://pubmed.ncbi.nlm.nih.gov/31147392/>; Ottenhoff et al., 2020, <https://pubmed.ncbi.nlm.nih.gov/32071072/>).

We show that cholesterol balance is altered in FXS astrocytes and associated with changes in lipidomics and pre-inflammatory factors, which may be rescued by some but not all statins.

In RESULTS, first paragraph: Reduced ABCA1 expression in human FXS astrocytes, there is sentence: “Similarly, human embryonic stem cell (hESC)-derived FMR1 KO astrocytes expressed less ABCA1 than their isogenic controls in RNA Seq analysis (Log2 -6.41-fold change, P=0.0004, P adjust = 0.125)²⁵. I read carefully published reference 25 and could not find any data about ABCA1 downregulation.

RESPONSE:

The RNA Seq data are introduced in the GLIA publication, but ABCA1 with p adjust = 0.125 as criterion p adjust < 0.1 was not listed in the publication. The RNA Seq data are available now at the Gene Expression Omnibus database under accession number GSE228378.

Figure 1B. ABCA1 immunofluorescence shows as nuclear stain; if ABCA1 is transporter, the staining should be membrane staining. There are cells that express ABCA1 but are not GFAP positive. ABCA1 Ab on Abcam website is not convincing image for IF.

RESPONSE:

We counted ABCA1 in all human astrocytes. Over 90% of astrocytes express GFAP varying in strength of expression level and show perinuclear and membrane staining of ABCA1. Our data show reduced expression of ABCA1 that recycles between the plasma membrane and endosomal compartments, a process involved in the transport of intracellular cholesterol into ApoE. The results suggested that this process is altered in FXS astrocytes and may thereby affect subcellular localization of ABCA1 immunoreactivity in astrocytes. However, our quantitative analysis of ABCA1 expression in human and mouse FXS and control astrocytes did not address changes in subcellular localization. We do see some nuclear staining in 10% of cells that don't express GFAP but this is a small number of cells that unlikely contributed to the observed differences in ABCA1 levels.

Figure 1D. Very confusing figure without sufficient description. The numbers below graph represent different cell lines used in a study – what does HEL mean? Does each bar represent one technical replicate? Conclusion: “total cholesterol content did not differ between FXS and control astrocyte conditioned medium”. “The concentration of cholesterol was determined in ACM of human FXS and control astrocytes using the cholesterol Quantification kit (Sigma-Aldrich) and fluorometric detection according to the manufacturer’s instructions”. This is not reliable method for cholesterol quantification. Especially when medium used in the experiment contains 10% FBS (mouse ACM was DMEM plus 10% FBS). In the excel supplementary info, the authors show experimental details for cholesterol fluorometric analysis and the numbers are % of control. What is % of control for the control cells?

RESPONSE:

The graphs in Figure 1D show cholesterol amount in ACM of human astrocytes derived from different iPSC lines, studied in the absence of FBS in Advanced DMEM/F12 containing al-

bumin-lipid supplement. The Quantification kit was used to analyze amount of total cholesterol under basal condition and after treatment with retinoic acid in astrocytes derived from different hiPSC lines in order to identify potential individual differences. The values were normalized to the average cholesterol content in control cell lines and the excel supplementary info is revised to show same values. The names of the cell lines (HEL code indicates that the cell line was produced at the University of Helsinki) are replaced to the figure legend in the revised manuscript. The results were supported by LC-MS analysis.

Figure 1E. Cholesterol level relative abundance measured by mass spec: description of mass spec cholesterol measurement is missing. Why are numbers 0 to 14,000? What does the dashed line with italic medium represent? Again, no difference in cholesterol level detected in ACM by mass spec between control and FXS.

RESPONSE:

The values were intensities normalized to total ion abundance of sample, but for consistency, we have now converted them to percentages of control; thus Ctrl -RA got the value 100. Human astrocytes were cultured without serum in Advanced DMEM/F12, which contained containing albumin-lipid supplement to support the cell growth. Medium without cells was included to the analysis of ACM using LC-MS. The dashed line represents medium cholesterol content without astrocytes, indicating that cholesterol was increased in medium.

Figure 1F. There is no information how is cholesteryl ester measured. There is no clear description of the Figure. Was alkyne hydrolysis used to remove esters? What does dotted line labeled with Medium shows?

RESPONSE:

Cholesteryl esters were measured by LC-MS as indicated in Figure Legend. In Methods, we explain that CE was determined as sum of several individual CE molecules which were identified based on match of retention time, accurate MS m/z value, and MS/MS spectra compared to external databases. Thus CEs were detected as molecular species without hydrolysis (mass spectrometric, not enzymatic colorimetric data). The dashed line represents cholesteryl content in medium without astrocytes, demonstrating slightly decreased content in the presence of FXS or control astrocytes.

Figure 1H. In results section authors wrote: "Similar to ABCA1 expression, SLC27A1 was reduced in FXS iPSC-derived astrocytes (Fig 1h) and seen in hESC-derived astrocytes lacking FMRP (log₂ -7.38-fold change, P=0.034) 25 when compared with controls". I read carefully published reference 25 and could not find any data about SLC27A1 or FATP1 down-regulation.

RESPONSE:

The RNA Seq data were used and the data are available at the Gene Expression Omnibus database under accession number GSE228378. In the present study, the reduced expression of ABCA1 and SLC27A1 were confirmed in iPSC-derived astrocytes derived from 3 control and 4 different cell lines. The reference indicating the RNA Seq data used for ABCA1 expression analysis has been replaced in the revised manuscript. The reduced SLC27A1 expression in the RNA Seq data has been omitted from the text for the avoidance of doubt and because the results were shown appropriately in biological replicates.

The golden standard units for presenting qPCR data and relative mRNA expression are delta delta Ct. The authors use relative mRNA expression and have units from 0 to 0.2 and 0 to 0.001. Based on the presentation it is impossible to understand the level of expression of ABCA1 and FAT1P in astrocytes. Is GAPDH the best normalizer? The ideal normalizer should be at the expression level similar to the actual gene of interest. It seems that GAPDH

is expressed at very high levels and the two genes of interest are present at extremely low levels.

RESPONSE:

We show ABCA1 and FAT1P relative mRNA delta delta Ct expression in the revised Figure and in the data summary. We have shown that GAPDH is not regulated in FXS astrocytes and suits well for normalization.

The authors should provide the accession number for sequences that were used to generate primers. Were the mouse primers used to amplify human genes? The primer sequences for ABCA1 and GAPDH published in Table 1 show primers that are specific to mice; not human.

RESPONSE:

The Figures show only PCR analysis of human astrocytes and the human primers are indicated appropriately.

Figure 2a. ABCA1 relative immunoreactivity: difference is really small and there is lack of description how is this measurement done. In the Supplementary Excel file there is list of images and WT % of control and Fmr1 KO % of control. What are controls for WT and what are controls for Fmr1 KO cells? If it is %, the graph in 2a shows scale 0-250% and the numbers in excel show 0.5 – 2.0.

RESPONSE:

Confocal images of cultured mouse astrocytes were taken with a confocal laser-scanning microscope for analysis of ABCA1 immunoreactivity. High-resolution optical sections (1024 × 1024 pixel format) were captured with a 20× zoom at 0.5 μm step intervals. At least 10 images were captured per culture (100-200 astrocytes) with 4-5 cultures per group. Using Image J, each z stack was collapsed into a single image by projection, and split by color. GFAP-expressing cells were outlined and saved in the ROI manager and used to measure area and mean intensity of ABCA1 immunoreactivity in GFAP-expressing cells and corrected by subtracting the background intensity. Average intensity of ABCA1 immunoreactivity was calculated for each image (10-20 cells). The data were normalized to the average value in the WT astrocytes from the same culture. The % values are revised to correlate in the Excel data sheet.

Figure 2b,c,d show relative abundance of cholesterol, cholesteryl ester and desmosterol in CM. This is very problematic measurement because the mouse astrocytes were grown in DMEM with 10% FBS. Was the cellular content measured? What is the baseline cholesterol level in the medium?

RESPONSE:

Cholesterol, cholesteryl ester and desmosterol were analyzed using LC-MS (and -MS/MS) in mouse ACM and astrocytes. Cholesterol level was not analyzed in the medium without astrocytes. Cholesterol is predominant sterol in the medium containing 10% FBS, which also contains desmosterol. FBS contains approximately 300Rg/ml cholesterol but fluctuates between different batches, and it would be difficult to correlate it with MS data. Although we are not able to provide an exact concentration, we analyzed levels of cholesterol in both medium and cells of WT and Fmr1 KO cultures, and are reporting relative changes in KO astrocytes as compared to WT astrocytes.

The conclusion on page 7 top paragraph: “The altered PC profile of Fmr1 KO ACM suggested that the absence of FMRP led to dysregulation of ABCA1-mediated efflux of PC along with cholesterol from Fmr1 KO astrocytes.” The data presented do not support this conclusion. The measurement of ABCA1 immunoreactivity is not the best method and the

changes presented are very small. Western blotting would be better choice compared to measuring fluorescence intensity on the microscope. While PC changes look convincing, the measurement of cholesterol is not ideal. Most of cholesterol measurement was done in medium containing 10% FBS and using a kit (and not mass spec). Even when mass spec was used, the authors did not use established standards to verify the identity of sterol peaks. The proper experiment would be feeding cells labeled precursors and precise quantification of labeled products in the medium. These experiments were not done.

RESPONSE:

We show reduced ABCA1 expression in both human and mouse astrocytes using both PCR and immunocytochemistry. Furthermore, the other results support altered ABCA1 function, based on the accumulation of cholesterol in mouse astrocytes and increased ApoE in human astrocytes shown in the revised manuscript. We tried to analyze ABCA1 with Western analysis but because of the large size and glycosylation of the protein, it was impossible to get reliable data, even using 18% PAGE for large proteins. We have revised Discussion to focus on the ABCA1-related results. All data generated with astrocytes cultured with serum were analyzed by MS using established standards and data bases. We have improved the method description of the identification methods of lipids utilizing match of retention time, accurate MS m/z value and MS/MS spectra compared to data bases, detailed in the methods. Thus, the identity of the sterols and other lipids that we report was fully confirmed.

Figure 5. Cholesterol biosynthesis starts with two molecules of acetyl CoA and formation of acetoacetyl-CoA. This is followed by a second condensation of acetyl CoA and acetoacetyl-CoA to form 3-hydroxy-3-methylglutaryl CoA (HMG-CoA) which is shown in the figure. The figure shows cholesterol precursors – these should be acetylCoA. Cholesterol immediate precursors are desmosterol and 7-dehydrocholesterol. The schematic shows elevated cholesterol in the cell, but the authors analyzed cholesterol in the medium. Desmosterol was found decreased in the mouse FXS astrocytes condition medium and elevated in cells. Desmosterol was not analyzed in human astrocytes. While it is important to summarize data this manuscript does not show the mechanisms involved in cholesterol homeostasis.

RESPONSE:

We have revised Figure 5 to focus on ABCA1/cholesterol related changes and to differentiate between mouse and human data with color codes.

References: marked in yellow are references that author used.

RESPONSE:

New references related to revised text have been added and renumbered.

REVIEWERS' COMMENTS:

Reviewer #4 (Remarks to the Author):

I have reviewed the manuscript and the authors responded to the reviewer's comments with the exception of Table listing the primer sequences. Authors wrote Human in front of the ABCA and GAPDH but did not change the actual sequence. The sequence shown is still mouse sequence.

Authors say: Human ABCA1: TCCTCTCCCAGAGCAAAAAGC; GTCCTTGGCAAAGTTCACAAATACT;
sequence shows mouse: XM_006537554.2=Mus musculus ABCA1 mRNA

Authors say: Human GAPDH: AACGACCCCTTCATTGAC; TCCACGACATACTCAGCAC
Sequence shows mouse: NM_001289726.2 = mus musculus GAPDH.

Responses to Reviewer:

The authors would like to thank the Reviewer for thoughtful and excellently thorough assessment of the manuscript. The manuscript has been revised to address the final concern as follow:

Reviewer #4

I have reviewed the manuscript and the authors responded to the reviewer's comments with the exception of Table listing the primer sequences. Authors wrote Human in front of the ABCA and GAPDH but did not change the actual sequence. The sequence shown is still mouse sequence. Authors say: Human ABCA1: TCCTCTCCCAGAGCAAAAAGC; GTCCTTGGCAAAGTTCACAAATACT; sequence shows mouse: XM_006537554.2=Mus musculus ABCA1 mRNA. Authors say: Human GAPDH: AACGACCCCTTCATTGAC; TCCACGACATACTCAGCAC Sequence shows mouse: NM_001289726.2 = mus musculus GAPDH.

- We really appreciate this important note, which led us to revise the human *ABCA1* and *GADPH* primer sequences in Table 1. The human *GADPH* primers used in the present study have been previously used in our studies of human iPSC-derived cells (Achuta et al. Sci. Signal. 11, 2018) and the human *ABCA1* primers were designed according to the studies of Kielar et al. (Kielar et al., Clinical Chemistry 47, 2089-97, 2001).